# DPsurv: Dual-Prototype Evidential Fusion for Uncertainty-Aware and Interpretable Whole Slide Image Survival Prediction

## Abstract

Pathology whole-slide images (WSIs) are widely used for cancer survival analysis because of their comprehensive histopathological information at both cellular and tissue levels, enabling quantitative, large-scale, and prognostically rich tumor feature analysis. However, most existing methods in WSI survival analysis struggle with limited interpretability and often overlook predictive uncertainty in heterogeneous slide images. In this paper, we propose DPsurv, a dual-prototype whole-slide image evidential fusion network that outputs uncertainty-aware survival intervals, while enabling interpretation of predictions through patch prototype assignment maps, component prototypes, and component-wise relative risk aggregation. Experiments on five publicly available datasets achieve the highest mean concordance index and the lowest mean integrated Brier score, validating the effectiveness and reliability of DPsurv. The interpretation of prediction results provides transparency at the feature, reasoning, and decision levels, thereby enhancing the trustworthiness and interpretability of DPsurv.

## 1 Introduction

Survival analysis, which predicts survival probabilities and outcomes over time, is a critical task in oncology for guiding therapeutic decision-making and improving patient outcomes. As a direct reflection of tumor progression, whole-slide images (WSIs) have recently emerged as an essential source of information for survival prediction in computational pathology (Zhang et al., 2025). The major challenges for identifying reliable prognostic patterns from WSIs lie in the gigapixel scale and the tissue heterogeneity (Wang et al., 2022; Xu et al., 2024). Failing to model and address these challenges can result in incomplete risk assessments, leading to suboptimal treatment planning and potentially compromised survival outcomes (Liu et al., 2025b; Shi et al., 2024b).

Deep neural networks (DNNs) have shown great promise in WSI survival analysis (Dimitriou et al., 2019) with their powerful feature extraction ability. However, given the extremely high resolution of WSIs, using DNNs for WSI feature extractions incurs substantial computational and annotation costs, rendering fully supervised learning impractical (Lu et al., 2021). Existing research has primarily focused on developing effective WSI representations for survival prediction with advanced weakly-supervised or unsupervised learning (Song et al., 2024). Multiple instance learning (MIL) is the most widely adopted approach, where instance-level features are extracted independently and aggregated via concatenation or attention (Maron & Lozano-Pérez, 1997). Patch clustering (Yu et al., 2023; Claudio Quiros et al., 2024) or prototype representation learning (Xu & Chen, 2023; Song et al., 2024) are popular in unsupervised WSI learning. While these advances mitigate the challenges of modeling super-resolution WSIs, tissue heterogeneity remains insufficiently addressed.

The inadequate modeling of tissue heterogeneity in WSIs directly undermines interpretability in survival analysis. Clinically, distinct tissue components such as tumor epithelium, stroma, and necrosis each carry independent prognostic value, and even within the same component, different morphological patterns can imply different survival outcomes (Travis et al., 2011). Therefore, detecting and interpreting the subtle or ambiguous regions of WSIs that often carry decisive prognostic value is critical for reliable survival prediction. Weakly-supervised learning usually employs attention scores to highlight influential regions (Shao et al., 2021; Xiang & Zhang, 2023), but the resulting heatmaps

convey only relative importance and lack clinical interpretability for survival risk assessment. Unsupervised learning approaches attempt to explain slide-level embedding aggregation through methods such as prototypes or K-means clustering (Vu et al., 2023), yet they offer limited feature-level interpretability. Nevertheless, neither paradigm achieves end-to-end interpretability in survival analysis, encompassing modeling, reasoning, and decision levels.

Another important but overlooked consequence of tissue heterogeneity is the uncertainty of survival results. The inherent heterogeneity of WSIs and the presence of incomplete event labels (censoring) introduce uncertainty in survival outcomes (Gomes et al., 2021; Davidov et al., 2025).Conventional methods output point survival estimates without conveying the appropriate uncertainty, leading to misleading treatment suggestions (Dolezal et al., 2022). Moreover, uncertainty research has been primarily focused on classification models (Abdar et al., 2021) using techniques such as Bayesian reasoning (Yufei et al., 2022), Monte Carlo simulation (Gal & Ghahramani, 2016), ensembling (Lakshminarayanan et al., 2017), or Subjective Logic (Jiang et al., 2025), while the exploration of uncertainty in regression models remains limited. Recently, the newly introduced Gaussian random fuzzy numbers (GFRNs) under the epistemic random fuzzy set (ERFS) theory enable the direct modeling of aleatory and epistemic uncertainty in regression models (Denœux, 2021; 2023b), and show promising performance in reasoning with noisy and censored survival data in the real line (Huang et al., 2024b; 2025).

Inspired by these insights, we propose DPsurv[1], a Dual-Prototype Evidential Fusion Network for interpretable and reliable WSI survival prediction. It encodes WSIs into deep component embeddings using a patch prototype-guided Gaussian mixture model (GMM) and maps them into an evidence space with component prototype-based GRFNs. The component-level evidence is then aggregated to estimate the lower and upper bounds of the survival function. The contributions are as follows:

- We propose a Dual-Prototype Evidential Fusion Network that addresses super-resolution and tissue heterogeneity challenges in WSI survival prediction, while providing prediction intervals that explicitly model aleatory and epistemic uncertainty.
- We demonstrate that DPsurv enables end-to-end interpretability in survival analysis, tracing pathways from deep embeddings to survival evidence and ultimately to component-level relative risk.
- We conduct extensive experiments and evaluations to assess the discriminative and calibration abilities of the model, and show that DPsurv achieves state-of-the-art performance.

## 2 RELATED WORK

**Learning approaches for WSI survival analysis** WSI-based survival prediction research can be broadly grouped into weakly-supervised and unsupervised methods. Weakly-supervised approaches are predominantly based on MIL, where patch features are generated by pre-trained feature extractors and then aggregated into a slide-level representation and mapped to survival risk via a prediction head (Yao et al., 2020; Xiang & Zhang, 2023). Based on the aggregation approaches, MIL approaches can be categorized into cluster (Zhou et al., 2024; Liu et al., 2025a), attention (Yang et al., 2024; Jiang et al., 2024; Kapse et al., 2024), and graph-based (Zheng et al., 2024; Li et al., 2024; Bui et al., 2024). Studies also extend MIL into multiscale modeling to learn hierarchical representations (Chen et al., 2022; Li et al., 2022; Deng et al., 2024). Unsupervised representation learning aims to construct explicit slide-level representations that preserve morphological heterogeneity in an unsupervised manner with strategies such as patch embedding clustering (Zaheer et al., 2017; Vu et al., 2023; Yu et al., 2023; Claudio Quiros et al., 2024), prototype learning (Mialon et al., 2020; Xu & Chen, 2023) and compact morphological prototype learning (Song et al., 2024).

**Uncertainty and interpretability in WSI survival analysis** Uncertainty studies in WSI survival analysis can be categorized into probabilistic and non-probabilistic methods (Huang et al., 2024a). Probabilistic approaches model uncertainty by relying on probability distributions. For example, Tang et al. (2025) estimates aleatoric uncertainty through a likelihood–based loss to provide fine-grained reliability scoring, Tang et al. (2023) injects aleatoric uncertainty into the Cox loss via a sample-dependent variance term, Yufei et al. (2022) applies a Bayesian formulation of attention weights to improve calibration. Non-probabilistic approaches model uncertainty without explicit probability distributions, often leveraging evidential or belief-based frameworks (Huang et al.,

---

[1] https://anonymous.4open.science/r/DPsurv-C06F

2024a). One representative method is Subjective Logic, which links evidential strength to the parameters of a Dirichlet distribution: Shi et al. (2024a) outputs Dirichlet evidence at the instance level and aggregates to bag-level predictions, Jiang et al. (2025) parameterizes survival predictions using Dirichlet evidence for calibration in multi-scale pathology–genomics fusion. Another direction employs evidential neural networks with GRFNs to capture survival uncertainty in both unimodal and multimodal settings (Huang et al., 2024c; 2025).

WSI survival interpretability studies have primarily been addressed within weakly-supervised MIL frameworks that use attention or pooling-related mechanisms to explain the aggregation process of slide embedding, such as local attention for patch-specific importance (Ilse et al., 2018), morphological prototypes (Yao et al., 2020), transformer modules for spatial correlations (Shao et al., 2021), multiscale embeddings with contrastive pretraining (Li et al., 2021), and low-rank attention for patch dependencies (Xiang & Zhang, 2023). Unsupervised approaches mainly focus on constructing morphology-associated slide representations for interpretation study through feature averaging, cluster counts, or optimal-transport/GMM-based prototypes (Zaheer et al., 2017; Mialon et al., 2020; Claudio Quiros et al., 2024; Yu et al., 2023; Vu et al., 2023; Song et al., 2024). However, these strategies often focus only on feature-level interpretability, thereby limiting model expressivity and transparency in the decision pathway from histological representation to survival outcomes.

# 3 METHOD

Figure 1 illustrates the overall DPsurv framework, which consists of deep slide component embedding, component evidence modeling, and component evidence mixture. We start with a preliminary introduction to regression uncertainty modeling and explain the proposed DPsurv and the optimization function.

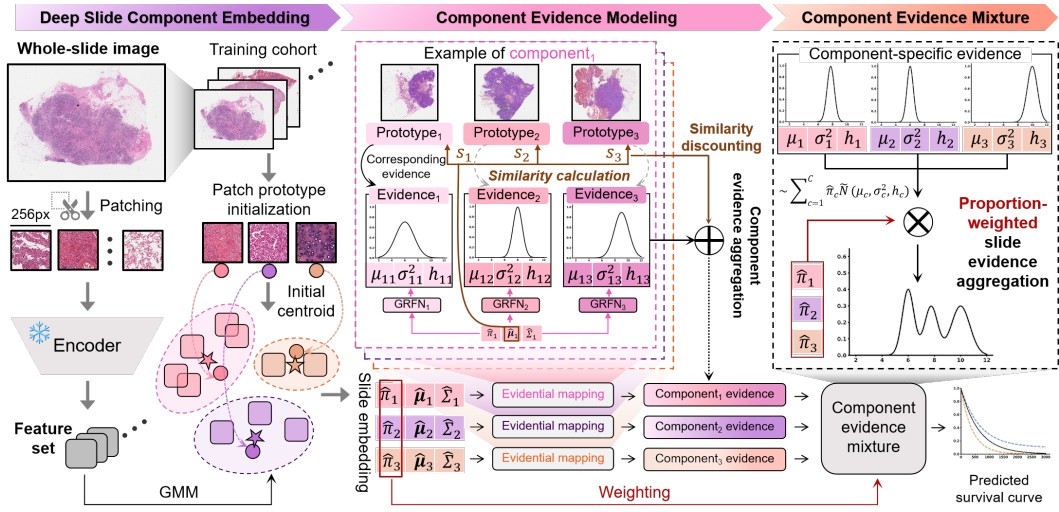

Figure 1: **Overview of the DPsurv framework** Deep Slide Component Embedding encodes WSI into deep feature embeddings with patch prototypes; Component Evidence Modeling maps deep embeddings into component evidence through component prototypes, and Component Evidence Mixture aggregates component evidence into transformed survival functions, illustrated by the Plausibility (blue dashed line) and Belief (orange dashed line) curves.

## 3.1 PRELIMINARIES

**Gaussian Random Fuzzy Numbers** In ERFS theory, a GRFN is a random fuzzy subset of the real line defined by the membership function $\varphi(x; M, h) = \exp\left(-\frac{h}{2}(x - M)^2\right)$, whose mode $M$ is a Gaussian random variable with $M \sim \mathcal{N}(\mu, \sigma^2)$ (Denœux, 2023b). A GRFN can be represented by $\tilde{Y} \sim \tilde{\mathcal{N}}(\mu, \sigma^2, h)$, where $\mu$ is the location parameter, and $\sigma$ and $h \in [0, +\infty)$ represent the aleatoric and epistemic uncertainties, respectively. Given a GRFN $\tilde{Y} \sim \tilde{\mathcal{N}}(\mu, \sigma^2, h)$, the degrees of

plausibility and belief for any interval $[x, y]$ can be calculated using:

$$Pl_{\tilde{Y}}([x,y]) = \Phi\left(\frac{y-\mu}{\sigma}\right) - \Phi\left(\frac{x-\mu}{\sigma}\right) + pl_{\tilde{Y}}(x)\,\Phi\left(\frac{x-\mu}{\sigma\sqrt{1+h\sigma^2}}\right) + pl_{\tilde{Y}}(y)\left[1 - \Phi\left(\frac{y-\mu}{\sigma\sqrt{1+h\sigma^2}}\right)\right], \quad \text{(1a)}$$

$$Bel_{\tilde{Y}}([x,y]) = \Phi\left(\frac{y-\mu}{\sigma}\right) - \Phi\left(\frac{x-\mu}{\sigma}\right) - pl_{\tilde{Y}}(x)\left[\Phi\left(\frac{(x+y)/2-\mu+(y-x)h\sigma^2/2}{\sigma\sqrt{1+h\sigma^2}}\right) - \Phi\left(\frac{x-\mu}{\sigma\sqrt{1+h\sigma^2}}\right)\right]$$

$$+ pl_{\tilde{Y}}(y)\left[\Phi\left(\frac{(x+y)/2-\mu-(y-x)h\sigma^2/2}{\sigma\sqrt{1+h\sigma^2}}\right) - \Phi\left(\frac{y-\mu}{\sigma\sqrt{1+h\sigma^2}}\right)\right], \quad \text{(1b)}$$

where $pl_{\tilde{Y}}(x) = \frac{1}{\sqrt{1+h\sigma^2}}\exp\left(-\frac{h(x-\mu)^2}{2(1+h\sigma^2)}\right)$ is the contour function and $\Phi$ denotes the standard normal cumulative distribution function. Further details are provided in Appendix A.1.

With GRFN, we can compute two types of prediction intervals. An $\alpha$-level belief prediction interval (BPI) is defined as $\mu \pm v$ such that $Bel_{\tilde{Y}}([\mu - v, \mu + v]) = \alpha$, thereby explicitly incorporating epistemic uncertainty. In contrast, An $\alpha$-level probabilistic prediction interval (PPI) is given by $\mu \pm \Phi^{-1}\left(\frac{1+\alpha}{2}\right)\sigma$, which relies only on Gaussian variance without accounting for epistemic uncertainty.

**From GRFNs to Survival prediction** Let $T \in (0, \infty)$ be a random variable denoting the survival time, the lower and upper bounds of the survival function can be modeled by $Bel_{\tilde{Y}}((\log t, \infty))$ and $Pl_{\tilde{Y}}((\log t, \infty))$, respectively, with which the true survival function $S(t) = p(T > t)$ satisfies

$$Bel_{\tilde{Y}}((\log t, \infty)) \leq S(t) \leq Pl_{\tilde{Y}}((\log t, \infty)), \quad \text{(2)}$$

where $Y = \log T$, a logarithmic transformation (Denœux, 2023b) that maps this positive random variable onto the real line and makes it compatible with the domain of GRFNs.

## 3.2 DEEP SLIDE COMPONENT EMBEDDING

Here, we introduce the deep slide component embedding using patch prototypes and Gaussian Mixture Model (GMM) (Dempster et al., 1977; Kim, 2022). Given the strong feature representation ability of foundation models, a WSI foundation model is used to map WSIs into patch-level embeddings, with each WSI for subject $i$ being segmented into non-overlapping patches $\mathbf{X}^i = \{\mathbf{x}_1^i, \ldots, \mathbf{x}_{N_i}^i\}$, $\mathbf{x}_n^i \in \mathbb{R}^{W \times H \times 3}$. Consequently, a set of patch embeddings for WSI is represented by $\mathbf{Z}^i = \{\mathbf{z}_1^i, \ldots, \mathbf{z}_{N_i}^i\}$ with $\mathbf{z}_n^i = f_{\text{enc}}(\mathbf{x}_n^i) \in \mathbb{R}^d$, $f_{\text{enc}}(\cdot)$ is the foundation model. However, these high-dimensional patch embeddings make survival analysis challenging. Following PANTHER (Song et al., 2024), we map the high-dimensional representation $\mathbf{Z}^i \in \mathbb{R}^{N_i \times d}$ into a low-dimensional embedding $\mathbf{z}_{\text{WSI}}^i \in \mathbb{R}^{C \times (2d+1)}$ while preserving essential morphological information using patch prototypes:

$$\mathbf{z}_{\text{WSI}}^i = \left[\sum_{n=1}^{N_i} \phi^i(\mathbf{z}_n^i, \mathbf{h}_1), \ldots, \sum_{n=1}^{N_i} \phi^i(\mathbf{z}_n^i, \mathbf{h}_C)\right], \quad \text{(3)}$$

where $\mathbf{h}_c \in \mathbb{R}^d$ is the patch prototype, and $\phi^i(\cdot, \cdot)$ is a similarity-based function that maps a patch embedding-prototype pair into a post-aggregation component embedding. GMM is then used to estimate $\phi^i(\cdot, \cdot)$ with the assumption that the patch embedding $\mathbf{z}_n^i$ is generated from a weighted sum of its conditional densities under each patch-prototype-aligned Gaussian component:

$$p(\mathbf{z}_n^i; \theta^i) = \sum_{c=1}^{C} p(c_n^i = c; \theta^i) \cdot p(\mathbf{z}_n^i | c_n^i = c; \theta^i) = \sum_{c=1}^{C} \pi_c^i \cdot \mathcal{N}(\mathbf{z}_n^i; \boldsymbol{\mu}_c^i, \Sigma_c^i), \quad \text{s.t.} \sum_{c=1}^{C} \pi_c^i = 1,$$

where $\theta^i = \left\{\pi_c^i, \boldsymbol{\mu}_c^i, \Sigma_c^i\right\}_{c=1}^{C}$ denote the GMM parameters (see Appendix A.3), $\pi_c^i$ is the prior probability that $\mathbf{z}_n^i$ originates from the $c$-th Gaussian component, for which $\boldsymbol{\mu}_c^i, \Sigma_c^i$ represent a morphological prototype and its variations for the $c$-th Gaussian component. Finally, WSI $i$ is represented by $\mathbf{z}_{\text{WSI}}^i \in \mathbb{R}^{C \times (2d+1)}$

$$\mathbf{z}_{\text{WSI}}^i = [\hat{\pi}_1^i, \hat{\boldsymbol{\mu}}_1^i, \hat{\Sigma}_1^i, \ldots, \hat{\pi}_C^i, \hat{\boldsymbol{\mu}}_C^i, \hat{\Sigma}_C^i]. \quad \text{(4)}$$

**Embeddings interpretation with assignment map** In the GMM framework, each embedding $\mathbf{z}_n^i$ is probabilistically associated with a set of prototypes, where the posterior responsibility of component

$c$ is determined by the mixture weight $\pi_c^i$ and its Gaussian likelihood. Each embedding is then assigned to the prototype $\mathbf{h}_{c_n^*}$ with the highest posterior responsibility, formally defined as

$$c_n^* = \arg \max_{c \in \{1,\ldots,C\}} \frac{\pi_c^i \, \mathcal{N}\big(\mathbf{z}_n^i; \, \boldsymbol{\mu}_c^i, \, \Sigma_c^i\big)}{\sum_{k=1}^C \pi_k^i \, \mathcal{N}\big(\mathbf{z}_n^i; \, \boldsymbol{\mu}_k^i, \, \Sigma_k^i\big)}. \tag{5}$$

The resulting assignment map reveals distinct morphological patterns across the WSI. Projecting these assignments back onto the WSI yields a patch prototype assignment map that highlights the spatial distribution of pathology-related visual concepts.

## 3.3 COMPONENT EVIDENCE MODELING

Here, we introduce the component evidence embedding modeling with component prototypes using GRFNs. Inspired by Huang et al. (2025), we map deep component embeddings $\mathbf{z}_{\text{WSI}}$ into evidence space via component prototypes using GRFNs. Let $\boldsymbol{p}_{c,1}, \ldots, \boldsymbol{p}_{c,K} \in \mathbb{R}^d$ denote $K$ prototype vectors for the $c^{\text{th}}$ Gaussian component $\mathbf{z}_{\text{WSI-c}} = [\hat{\pi}_c, \hat{\boldsymbol{\mu}}_c, \hat{\Sigma}_c]$. The evidence contributed by component prototype $\boldsymbol{p}_{c,k}$ for $\mathbf{z}_{\text{WSI-c}}$ is represented by a GRFN

$$\tilde{Y}_{c,k} \sim \tilde{\mathcal{N}}(\mu_{c,k}, \sigma_{c,k}^2, h_{c,k}), \tag{6}$$

where $\sigma_{c,k}^2$ and $h_{c,k}$ denote the variance and precision of prototype $\boldsymbol{p}_{c,k}$, and the mean is parameterized as $\mu_{c,k} = \boldsymbol{\beta}_{c,k}^\top \mathbf{z}_{\text{WSI-c}} + \beta_{c,k0}$, with $\boldsymbol{\beta}_{c,k} \in \mathbb{R}^{2d+1}$ the coefficient vector and $\beta_{c,k0} \in \mathbb{R}$ a scalar. The similarity between Gaussian component $\mathbf{z}_{\text{WSI-c}}$ and prototype $\boldsymbol{p}_{c,k}$ is measured using

$$s_{c,k} = \exp\big[-\gamma_{c,k}^2 \, d_{\cos}(\hat{\boldsymbol{\mu}}_c, \boldsymbol{p}_{c,k})\big], \tag{7}$$

where $d_{\cos}\big(\hat{\boldsymbol{\mu}}_c, \boldsymbol{p}_{c,k}\big) = \frac{1}{2}\Big(1 - \frac{\hat{\boldsymbol{\mu}}_c^\top \boldsymbol{p}_{c,k}}{\|\hat{\boldsymbol{\mu}}_c\| \|\boldsymbol{p}_{c,k}\|}\Big)$ denotes the cosine distance, and $\gamma_{c,k} > 0$ is a positive scalar that controls the rate of decay with distance. The evidence of the $c^{\text{th}}$ Gaussian component $\tilde{Y}_c \sim \tilde{\mathcal{N}}\big(\mu_c, \sigma_c^2, h_c\big)$ is obtained by aggregating the evidence of its component prototypes $\{\tilde{Y}_{c,k}\}_{k=1}^K$ using the unnormalized product–intersection rule

$$\mu_c = \frac{\sum_{k=1}^K s_{c,k} \, h_{c,k} \, \mu_{c,k}}{\sum_{k=1}^K s_{c,k} \, h_{c,k}}, \quad \sigma_c^2 = \frac{\sum_{k=1}^K s_{c,k}^2 \, h_{c,k}^2 \, \sigma_{c,k}^2}{\big(\sum_{k=1}^K s_{c,k} \, h_{c,k}\big)^2}, \quad h_c = \sum_{k=1}^K s_{c,k} \, h_{c,k}. \tag{8}$$

**Survival evidence interpretation with component prototypes** To characterize the role of component prototypes in survival evidence, we retrieve representative training samples belonging to the same component that exhibit the highest cosine similarity to each component prototype, thereby providing interpretable pathological characterizations. For each of the $c^{\text{th}}$ Gaussian component and its component prototype $\boldsymbol{p}_{c,k}$, we assign $\exp(\mu_c)$ and $\exp(\mu_{c,k})$ as the most plausible survival times (PST), which serve as quantitative indicators of the associated risk evidence. In summary, the survival evidence of a component is derived through a similarity-based aggregation of the risk evidence contributed by its component prototypes.

## 3.4 COMPONENT EVIDENCE MIXTURE

We then explain the mixture of component evidence for survival prediction using the evidence mixture mechanism (Denœux, 2023a). Let $W$ be a random variable taking values in $\{1, \ldots, C\}$, we reformulate slide embedding as

$$\mathbf{z}_{\text{WSI}} = \sum_{c=1}^C \mathbf{1}_{\{W=c\}} \cdot [\hat{\boldsymbol{\mu}}_c, \hat{\Sigma}_c], \tag{9}$$

where $P(W = c) = \hat{\pi}_c$ denotes the prior probability that a patch embedding belongs to the $c^{\text{th}}$ Gaussian component. Accordingly, $\tilde{Y}_c$ is a conditional GRFN given $W = c$ and the slide-level evidence is a mixture GRFN (m-GRFN, Appendix A.4), denoted by $\tilde{Y} \sim \sum_{c=1}^C \hat{\pi}_c \tilde{\mathcal{N}}(\mu_c, \sigma_c^2, h_c)$. The degrees of belief and plausibility induced by m-GRFN $\tilde{Y}$ are the weighted sums of the belief and plausibility functions of its mixture components, given by

$$Bel_{\tilde{Y}}([x,y]) = \sum_{c=1}^C \pi_c \, Bel_{\tilde{Y}_c}([x,y]), \quad Pl_{\tilde{Y}}([x,y]) = \sum_{c=1}^C \pi_c \, Pl_{\tilde{Y}_c}([x,y]). \tag{10}$$

Letting $y \to \infty$, the degrees of belief and plausibility induced by m-GRFN $\tilde{Y}$ correspond to the lower and upper bounds of survival functions:

$$Bel_{\tilde{Y}}([x, \infty)) = 1 - \Phi\left(\frac{x - \mu}{\sigma}\right) - pl_{\tilde{Y}}(x) + pl_{\tilde{Y}}(x)\,\Phi\left(\frac{x - \mu}{\sigma\sqrt{1 + h\sigma^2}}\right), \tag{11a}$$

$$Pl_{\tilde{Y}}([x, \infty)) = 1 - \Phi\left(\frac{x - \mu}{\sigma}\right) + pl_{\tilde{Y}}(x)\,\Phi\left(\frac{x - \mu}{\sigma\sqrt{1 + h\sigma^2}}\right). \tag{11b}$$

For each WSI $i$, the survival function at time $t$ is computed as:

$$S_i(t) = \lambda\, Bel_{\tilde{Y}^i}\big([\log t, \infty)\big) + (1 - \lambda)\, Pl_{\tilde{Y}^i}\big([\log t, \infty)\big), \tag{12}$$

where $\lambda \in [0, 1]$ balances the impact of belief and plausibility. Find more details in Appendix A.2.

**Survival prediction interpretation with Component-wise Relative Risk** Given an m-GRFN, we introduce a relative risk measure to quantify the contribution of component-specific evidence to the final survival outcome. In a GRFN, $\mu$ denotes the plausible log-survival time and is inversely related to risk. Accordingly, for a WSI, the component-wise relative risk is defined as

$$r_c = 1 - \frac{\mu_c - \min_j \mu_j}{\max_j \mu_j - \min_j \mu_j}, \quad c = 1, \ldots, C. \tag{13}$$

This formulation enables the visualization of relative risk distributions across WSIs, providing spatial interpretability at the tissue level.

### 3.5 MIXTURE EVIDENTIAL LOSS

We propose a *mixture evidential loss* for survival prediction under uncertainty, which integrates mixture-based evidence with the negative log-likelihood loss (Zadeh & Schmid, 2020). This formulation links uncertainty with survival probability while addressing censored–uncensored weighting. We first partition uncensored survival times in the training set into $B$ quantile-based bins, denoted as $b_j = [T_j, T_{j+1})$, such that each bin contains the same number of uncensored samples. We then calculate the negative log-likelihood of uncensored and all subjects, respectively, with

$$\ell_i^{\mathrm{unc}} = -(1 - c_i) \sum_{j=1}^{B} \mathbf{1}_{\{y_i \in b_j\}} \log\big(S_i(T_j) - S_i(T_{j+1})\big), \tag{14a}$$

$$\ell_i = \ell_i^{\mathrm{unc}} - c_i \sum_{j=1}^{B} \mathbf{1}_{\{y_i \in b_j\}} \log S_i(T_{j+1}), \tag{14b}$$

where $c_i$ is the censoring indicator. The *mixture evidential loss* is then defined as

$$\mathcal{L}_{\mathrm{Mix}} = \frac{1}{N} \sum_{i=1}^{N} \left[(1 - \alpha)\,\ell_i + \alpha\,\ell_i^{\mathrm{unc}}\right] + \xi\,\mathcal{R}_1 + \rho\,\mathcal{R}_2, \tag{15}$$

where $\alpha \in [0, 1]$ is a trade-off parameter that balances between censored and uncensored likelihood contributions, thus controlling the robustness of the objective. The terms $\mathcal{R}_1 = \frac{1}{NCK} \sum_{i=1}^{N} \sum_{c=1}^{C} \sum_{k=1}^{K} h_{c,k}^i$ and $\mathcal{R}_2 = \frac{1}{NCK} \sum_{i=1}^{N} \sum_{c=1}^{C} \sum_{k=1}^{K} \big(\gamma_{c,k}^i\big)^2$ are regularization penalties that encourage stable evidential learning, controlled by hyperparameters $\xi$ and $\rho$.

## 4 EXPERIMENTS

### 4.1 EXPERIMENTAL SETUP

**Datasets** Five cancers provided by TCGA are tested: Breast Invasive Carcinoma (BRCA), Bladder Urothelial Carcinoma (BLCA), Uterine Corpus Endometrial Carcinoma (UCEC), Kidney Renal Clear Cell Carcinoma (KIRC), and Lung Adenocarcinoma (LUAD). Following Song et al. (2024), we use 5-fold site-stratified cross-validation to minimize distribution differences between the training and test sets (Howard et al., 2021). Further dataset details are provided in Appendix B.1.

**Baselines** Methods without uncertainty-awareness (UA) are: ABMIL (Ilse et al., 2018), TransMIL (Shao et al., 2021), AttnMISL (Yao et al., 2020), ILRA (Xiang & Zhang, 2023) and PANTHER (Song et al., 2024). Methods with UA are: EVREG (Huang et al., 2025), UMSA (Jiang et al., 2025) and BayesMIL (Yufei et al., 2022). UNI2-h (Chen et al., 2024), pre-trained on a large-scale internal histology dataset, was used as the feature extractor for all comparison methods in this paper. Implementation details of all baselines are given in Appendix B.2.

**Evaluation Metrics** We use the Concordance index (C-index) (Harrell et al., 1982) to assess discrimination and use the integrated Brier score (IBS) and integrated (negative) binomial log-likelihood (IBLL) (Graf et al., 1999) to evaluate calibration (See Appendix B.3).

## 4.2 SURVIVAL ACCURACY

As shown in Table 1, DPsurv outperforms most existing baselines and demonstrates superior discriminative ability across cancer types. Specifically, it achieves the highest mean C-index across the five TCGA cohorts (0.685), ranking first on BLCA (0.652), LUAD (0.634), UCEC (0.719), and KIRC (0.739), while remaining competitive on BRCA. The relatively lower performance on BRCA may be attributed to high tumor heterogeneity and censoring rate, both of which hinder the learning of reliable component prototypes. For cross-cancer evaluation, all C-index values obtained by DPsurv exceed 0.63, demonstrating consistent performance across diverse cancer types, indicating that the learned component prototypes capture generalizable tissue features from heterogeneous histologies and effectively mitigate the influence of redundant information during evidence aggregation.

Table 1: **C-index** from 5-fold cross-validation on five TCGA datasets, with the averaged C-index across datasets (Avg). Best results are in **bold**; second best are underlined.

| | Method | BRCA | BLCA | LUAD | UCEC | KIRC | Avg (↑) |
|---|---|---|---|---|---|---|---|
| **✗ UA** | ABMIL | 0.656 (±0.05) | 0.555 (±0.05) | 0.631 (±0.09) | 0.619 (±0.10) | 0.644 (±0.09) | 0.621 |
| | TransMIL | 0.576 (±0.09) | 0.561 (±0.09) | 0.612 (±0.12) | 0.660 (±0.09) | 0.725 (±0.08) | 0.627 |
| | DSMIL | 0.638 (±0.02) | 0.586 (±0.03) | 0.615 (±0.07) | 0.697 (±0.11) | 0.700 (±0.08) | 0.647 |
| | AttnMISL | 0.652 (±0.05) | 0.487 (±0.10) | 0.583 (±0.06) | 0.642 (±0.10) | 0.681 (±0.05) | 0.609 |
| | ILRA | 0.584 (±0.07) | 0.575 (±0.07) | 0.585 (±0.03) | 0.681 (±0.07) | 0.683 (±0.10) | 0.622 |
| | PANTHER | **0.721** (±0.07) | 0.650 (±0.06) | 0.560 (±0.06) | 0.713 (±0.03) | 0.693 (±0.08) | 0.667 |
| **✓ UA** | EVREG | 0.646 (±0.09) | 0.599 (±0.09) | 0.577 (±0.08) | 0.668 (±0.11) | 0.618 (±0.10) | 0.622 |
| | UMSA | 0.673 (±0.06) | 0.565 (±0.07) | 0.626 (±0.11) | 0.660 (±0.09) | 0.646 (±0.10) | 0.634 |
| | BayesMIL | 0.678 (±0.09) | 0.611 (±0.07) | 0.604 (±0.10) | 0.716 (±0.08) | 0.695 (±0.09) | 0.661 |
| | DPsurv (ours) | 0.680 (±0.07) | **0.652** (±0.03) | **0.634** (±0.17) | **0.719** (±0.09) | **0.739** (±0.10) | **0.685** |

## 4.3 SURVIVAL INTERPRETABILITY

**Feature-phenotypes interpretation** We visualize the assignment map together with the patch prototype distribution to examine the morphological phenotypes captured in WSIs. As shown in Figure 2A with annotations from board-certified pathologists, the learned patch prototypes correspond to distinct morphological phenotypes, including tumor regions with varying cellular density, necrotic and inflammatory areas, and surrounding normal lung tissue and stromal regions.

**Survival reasoning interpretation** Figure 2B visualizes the reasoning process from each component to its corresponding evidence. For a given target component, its risk evidence is derived by combining the evidence from component prototypes according to their similarity. Pathologists' assessments confirm that the component prototypes identified by our model correspond to distinct phenotypic subtypes, such as solid sheets of tumor cells with minimal stroma, solid–acinar patterns with stromal plasma cell infiltrates, and acinar–cribriform growth with prominent lymphoid infiltration. Necrosis and mitotic activity are most evident in component prototype-1, less pronounced in component prototype-2, and least apparent in component prototype-3, which is consistent with the relative risk indicated by their predicted PST values. Moreover, the aggregation process is inherently

interpretable, as each component prototype's contribution can be examined via its PST and associated BPI. Aggregating evidence across component prototypes yields more stable risk estimates, reflected in narrower BPIs.

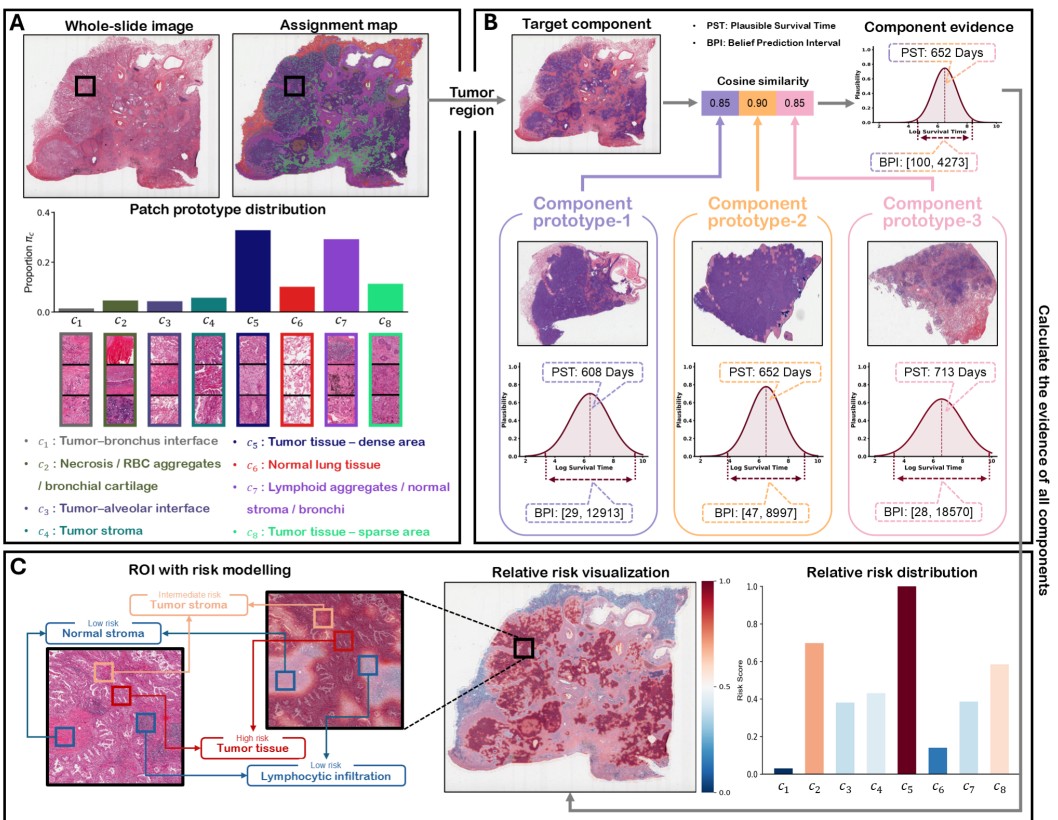

Figure 2: **Interpretation of the DPsurv in WSI survival prediction** (A) Visualization of the assignment map, prototype distribution, and morphological annotations provided by a board-certified pathologist. (B) Visualization of component prototypes and the reasoning process for component evidence modeling. (C) Decision making with component-wise relative risk and its distribution over the WSI and region of interest (ROI).

**Survival decision interpretation** Figure 2C illustrates how the decision-making process can be interpreted within the DPsurv framework. By mapping components to their corresponding risk evidence, we can visualize the relative risk distribution within a WSI. This enables the model to perform risk assessment and assign each component a reasonable risk level. For example, tumor regions are identified as the highest-risk areas, a finding that was confirmed by pathologists. Furthermore, by zooming into ROIs, we can link distinct morphological patterns to their associated risks, thereby facilitating a deeper understanding of the relationship between histologic characteristics and risk.

**Comparison with attention map** While attention heatmaps offer some interpretability by indicating which patches receive higher weights in the decision process, they remain limited to the feature level. In contrast, DPsurv provides a more structured and clinically meaningful interpretability framework, moving beyond patch-level weighting to deliver multi-level, pathology-informed, and quantitatively grounded explanations that better align with clinical reasoning and decision-making.

## 4.4 SURVIVAL UNCERTAINTY

Table 2 shows the uncertainty quantification performance. Among the compared methods, DPsurv achieves the lowest mean IBS (0.288) across the five TCGA cohorts, consistently outperforming all baselines. This demonstrates that DPsurv not only discriminates between high- and low-risk patients but also provides well-calibrated survival estimates across diverse cancer types. Notably,

all UA methods achieve mean IBS values below 0.7 (ranging from 0.288 to 0.697), whereas non-UA methods often exceed this threshold, with averages ranging from 0.523 to 0.920. This highlights the importance of explicitly modeling predictive uncertainty in survival analysis.

Table 2: **IBS** from 5-fold cross-validation on five TCGA datasets, with the averaged IBS across datasets (Avg). Best results are in **bold**; second best are underlined.

| | Method | BRCA | BLCA | LUAD | UCEC | KIRC | Avg (↓) |
|---|---|---|---|---|---|---|---|
| **✗ UA** | ABMIL | 0.788 (±0.19) | 0.651 (±0.10) | 0.697 (±0.25) | 0.870 (±0.12) | 0.609 (±0.23) | 0.723 |
| | TransMIL | 0.984 (±0.02) | 0.935 (±0.04) | 0.904 (±0.09) | 0.956 (±0.05) | 0.820 (±0.16) | 0.920 |
| | DSMIL | 0.731 (±0.21) | 0.470 (±0.10) | 0.527 (±0.12) | 0.694 (±0.18) | 0.525 (±0.20) | 0.589 |
| | AttnMISL | 0.946 (±0.06) | 0.647 (±0.20) | 0.757 (±0.10) | 0.909 (±0.04) | 0.678 (±0.05) | 0.788 |
| | ILRA | 0.985 (±0.02) | 0.936 (±0.04) | 0.875 (±0.19) | 0.904 (±0.10) | 0.857 (±0.09) | 0.911 |
| | PANTHER | 0.683 (±0.08) | 0.376 (±0.04) | **0.418** (±0.02) | 0.680 (±0.03) | 0.457 (±0.11) | 0.523 |
| **✓ UA** | EVREG | 0.569 (±0.09) | 0.450 (±0.08) | 0.468 (±0.05) | 0.562 (±0.08) | 0.542 (±0.07) | 0.518 |
| | UMSA | 0.737 (±0.26) | 0.597 (±0.16) | 0.697 (±0.26) | 0.851 (±0.20) | 0.600 (±0.20) | 0.697 |
| | BayesMIL | 0.746 (±0.19) | 0.538 (±0.09) | 0.644 (±0.14) | 0.815 (±0.09) | 0.554 (±0.21) | 0.659 |
| | DPsurv (ours) | **0.186** (±0.04) | **0.296** (±0.10) | 0.458 (±0.06) | **0.244** (±0.06) | **0.257** (±0.05) | **0.288** |

We have two observations in Figure 3. (1) BPIs achieve better calibration. A model is considered well calibrated if the coverage of its prediction intervals aligns with the nominal confidence level, corresponding to curves lying along the diagonal. In this ideal case, for any chosen $\alpha$, exactly $\alpha$ proportion of true survival times fall within the predicted intervals. We find that, compared to PPIs which tend to fall below the diagonal, BPIs lie closer to the diagonal across all datasets, reflecting more calibrated and precise survival predictions. These results demonstrate that explicitly accounting for epistemic uncertainty plays a critical role in achieving well-calibrated survival predictions. (2) BPIs exhibit conservative predictions. In most cases, BPIs lie slightly above the diagonal, suggesting that the model tends to produce conservative predictions that avoid underestimating risk. This conservative tendency is particularly desirable in survival analysis, as underestimating patient risk could lead to inappropriate treatment decisions, whereas slight overestimation ensures safer and more cautious clinical decision-making.

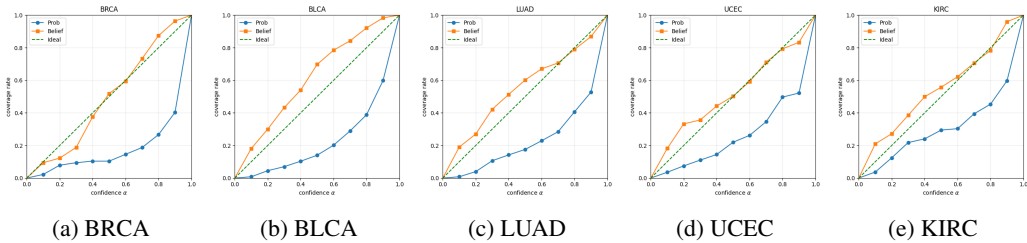

(a) BRCA      (b) BLCA      (c) LUAD      (d) UCEC      (e) KIRC

Figure 3: Calibration plots of DPsurv across five TCGA datasets, showing the proportion of $\alpha$-level BPIs and PPIs that contain the uncensored survival times, for $\alpha \in \{0.1, \ldots, 0.9\}$.

## 5 CONCLUSION

We propose DPsurv, a Dual-Prototype Evidential Fusion Network for WSI survival prediction. To enhance interpretability and uncertainty awareness, our framework integrates three key parts: i) patch prototype–based GMMs to derive slide-level component embeddings, ii) component prototype–based evidential modeling to map component embeddings into component-wise evidence, and iii) evidence mixture mechanism to fuse component-wise evidence into final predictive evidence. DPsurv achieves state-of-the-art discriminative and well-calibrated performance across five public datasets. Beyond performance, the model provides multi-level transparency and explicit uncertainty quantification, thereby offering a novel perspective for interpretable and uncertainty-aware survival prediction in computational pathology.

## ETHICS STATEMENT

This study adheres to the ICLR Code of Ethics. Our work does not involve human subjects, sensitive personal data, or potential harmful applications. The analysis is performed on publicly available TCGA datasets under the corresponding data usage policies. We explicitly confirm compliance with ethical standards and research integrity.

## REPRODUCIBILITY STATEMENT

We have taken significant steps to ensure reproducibility. All datasets used in this work are publicly available TCGA cohorts. Model architectures are detailed in Section 3. Proofs of theoretical results are provided in Appendix A. Detailed experimental settings, including hyperparameter configurations, are presented in Appendix B. The source code is available at `https://anonymous.4open.science/r/DPsurv-C06F`.

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

APPENDIX

## A  PROOF

### A.1  CONTOUR, PLAUSIBILITY AND BELIEF FUNCTION

**Derivation of the contour function.**  Starting from the definition,

$$pl_{\tilde{Y}}(x) = \mathbb{E}_M\big[\varphi(x; M, h)\big] = \int_{-\infty}^{+\infty} \varphi(x; m, h)\,\phi(m; \mu, \sigma)\,dm$$

$$= \frac{1}{\sigma\sqrt{2\pi}} \int_{-\infty}^{+\infty} \exp\!\big(-\tfrac{h}{2}(x-m)^2\big) \exp\!\Big(-\tfrac{(m-\mu)^2}{2\sigma^2}\Big)\,dm. \tag{16}$$

Following Denœux (2023b), the integrand can be factorized as

$$\exp\!\Big(-\frac{(m-\mu_0)^2}{2\sigma_0^2}\Big)\,\exp\!\Big(-\frac{h(x-\mu)^2}{2(1+h\sigma^2)}\Big), \tag{17}$$

with

$$\mu_0 = \frac{x\,h\sigma^2 + \mu}{1 + h\sigma^2}, \qquad \sigma_0^2 = \frac{\sigma^2}{1 + h\sigma^2}. \tag{18}$$

Evaluating the Gaussian integral then yields

$$pl_{\tilde{Y}}(x) = \frac{1}{\sigma\sqrt{2\pi}} \exp\!\Big(-\frac{h(x-\mu)^2}{2(1+h\sigma^2)}\Big) \int_{-\infty}^{+\infty} \exp\!\Big(-\frac{(m-\mu_0)^2}{2\sigma_0^2}\Big)\,dm$$

$$= \frac{1}{\sqrt{1+h\sigma^2}} \exp\!\Big(-\frac{h(x-\mu)^2}{2(1+h\sigma^2)}\Big). \tag{19}$$

**Derivation of the plausibility function.**  Assume $h > 0$. By the definition of plausibility over an interval,

$$\mathrm{Pl}_{\tilde{Y}}([x, y]) = P(M \le x)\,\mathbb{E}\big[\varphi(x; M, h) \mid M \le x\big] + P(x < M \le y) \cdot 1$$

$$+ P(M > y)\,\mathbb{E}\big[\varphi(y; M, h) \mid M > y\big]$$

$$= \Phi\big(\tfrac{x-\mu}{\sigma}\big)\,\mathbb{E}\big[\varphi(x; M, h) \mid M \le x\big] + \Big(\Phi\big(\tfrac{y-\mu}{\sigma}\big) - \Phi\big(\tfrac{x-\mu}{\sigma}\big)\Big)$$

$$+ \Big(1 - \Phi\big(\tfrac{y-\mu}{\sigma}\big)\Big)\mathbb{E}\big[\varphi(y; M, h) \mid M > y\big]. \tag{20}$$

Since $M \mid M \le x$ is a Gaussian truncated to $(-\infty, x]$ with probability density function

$$f(m) = \frac{1}{\sigma\sqrt{2\pi}\,\Phi\big(\tfrac{x-\mu}{\sigma}\big)} \exp\!\Big(-\tfrac{(m-\mu)^2}{2\sigma^2}\Big)\,\mathbf{1}_{(-\infty, x]}(m), \tag{21}$$

we get

$$\mathbb{E}\big[\varphi(x; M, h) \mid M \le x\big] = \frac{1}{\sigma\sqrt{2\pi}\,\Phi\big(\tfrac{x-\mu}{\sigma}\big)} \int_{-\infty}^{x} \exp\!\big(-\tfrac{h}{2}(x-m)^2\big) \exp\!\Big(-\tfrac{(m-\mu)^2}{2\sigma^2}\Big)\,dm. \tag{22}$$

Direct evaluation provides a closed form in terms of $\Phi(\cdot)$ and the previously derived $pl_{\tilde{Y}}(x)$; an analogous computation applies to $\mathbb{E}\big[\varphi(y; M, h) \mid M > y\big]$. Substituting both expressions into equation 20 gives the stated formula for $\mathrm{Pl}_{\tilde{Y}}([x, y])$.

**Derivation of the belief function.**  By duality between plausibility and belief,

$$\mathrm{Bel}_{\tilde{Y}}([x, y]) = 1 - \mathrm{Pl}_{\tilde{Y}}\big((-\infty, x] \cup [y, +\infty)\big). \tag{23}$$

Using the same decomposition as for plausibility, we write

$$P_{I_{\tilde{Y}}}\big((-\infty, x] \cup [y, +\infty)\big) = P(M \leq x) \cdot 1$$

$$+ P\big(x < M \leq \tfrac{x+y}{2}\big)\, \mathbb{E}\big[\varphi(x; M, h) \mid x < M \leq \tfrac{x+y}{2}\big]$$

$$+ P\big(\tfrac{x+y}{2} < M \leq y\big)\, \mathbb{E}\big[\varphi(y; M, h) \mid \tfrac{x+y}{2} < M \leq y\big]$$

$$+ P(M > y) \cdot 1$$

$$= \Phi\big(\tfrac{x-\mu}{\sigma}\big)$$

$$+ \left[ \Phi\Big(\tfrac{(x+y)/2-\mu}{\sigma}\Big) - \Phi\big(\tfrac{x-\mu}{\sigma}\big) \right] \mathbb{E}\big[\varphi(x; M, h) \mid x < M \leq \tfrac{x+y}{2}\big]$$

$$+ \left[ \Phi\big(\tfrac{y-\mu}{\sigma}\big) - \Phi\Big(\tfrac{(x+y)/2-\mu}{\sigma}\Big) \right] \mathbb{E}\big[\varphi(y; M, h) \mid \tfrac{x+y}{2} < M \leq y\big]$$

$$+ 1 - \Phi\big(\tfrac{y-\mu}{\sigma}\big). \tag{24}$$

On $x < M \leq (x+y)/2$, $M$ follows a Gaussian truncated to $(x, (x+y)/2]$ with probability density function

$$f(m) = \frac{1}{\sigma\sqrt{2\pi}\left[\Phi(\tfrac{(x+y)/2-\mu}{\sigma}) - \Phi(\tfrac{x-\mu}{\sigma})\right]} \exp\Big(-\tfrac{(m-\mu)^2}{2\sigma^2}\Big)\, \mathbf{1}_{(x,(x+y)/2]}(m). \tag{25}$$

Therefore,

$$\mathbb{E}\big[\varphi(x; M, h) \mid x < M \leq \tfrac{x+y}{2}\big] = \frac{\int_x^{(x+y)/2} \exp\big(-\tfrac{h}{2}(x-m)^2\big) \exp\Big(-\tfrac{(m-\mu)^2}{2\sigma^2}\Big)\, dm}{\sigma\sqrt{2\pi}\left[\Phi\Big(\tfrac{(x+y)/2-\mu}{\sigma}\Big) - \Phi\big(\tfrac{x-\mu}{\sigma}\big)\right]}$$

$$= \frac{\Phi\Big(\tfrac{(x+y)/2-\mu+h\sigma^2(y-x)/2}{\sigma\sqrt{h\sigma^2+1}}\Big) - \Phi\Big(\tfrac{x-\mu}{\sigma\sqrt{h\sigma^2+1}}\Big)}{\Phi\Big(\tfrac{(x+y)/2-\mu}{\sigma}\Big) - \Phi\big(\tfrac{x-\mu}{\sigma}\big)}\, pl_{\tilde{Y}}(x). \tag{26}$$

Similarly,

$$\mathbb{E}\big[\varphi(y; M, h) \mid \tfrac{x+y}{2} < M \leq y\big] = \frac{\Phi\Big(\tfrac{y-\mu}{\sigma\sqrt{h\sigma^2+1}}\Big) - \Phi\Big(\tfrac{(x+y)/2-\mu-(y-x)h\sigma^2/2}{\sigma\sqrt{h\sigma^2+1}}\Big)}{\Phi\big(\tfrac{y-\mu}{\sigma}\big) - \Phi\big(\tfrac{(x+y)/2-\mu}{\sigma}\big)}\, pl_{\tilde{Y}}(y). \tag{27}$$

Plugging these into equation 23 and equation 24 produces the closed-form expression of $\mathrm{Bel}_{\tilde{Y}}([x,y])$.

## A.2 MIXTURE OF GAUSSIAN RANDOM FUZZY NUMBER

By the definition of plausibility, $Pl_{\tilde{Y}}([x,y])$ can be expressed as

$$Pl_{\tilde{Y}}([x,y]) = \mathbb{E}_{M,W}\left[ \sup_{u \in [x,y]} \varphi\left(u, M, \prod_{c=1}^C h_c^{I(W=c)}\right) \right]$$

$$= \mathbb{E}_W \mathbb{E}_{M|W}\left[ \sup_{u \in [x,y]} \varphi\left(u, M, \prod_{c=1}^C h_c^{I(W=c)}\right) \right]$$

$$= \sum_{c=1}^C \pi_c\, \mathbb{E}_{M|W}\left[ \sup_{u \in [x,y]} \varphi(u, M, h_c) \,\Big|\, W = c \right]$$

$$= \sum_{c=1}^C \pi_c\, Pl_{\tilde{Y}_c}([x,y]). \tag{28}$$

Here, $Pl_{\tilde{Y}}([x,y])$ denotes the plausibility function of the mixture GRFN. By duality, the belief function $Bel_{\tilde{Y}}([x,y])$ can be derived in the same way, and is interpreted as the belief function associated with the same mixture GRFN.

### A.3 GAUSSIAN MIXTURE MODEL ESTIMATION DETAILS

For completeness, we provide the detailed derivation of the parameter estimation for the Gaussian Mixture Model described in Section 3.2.

We clarify that $\phi^i(\cdot,\cdot) : \mathbb{R}^d \times \mathbb{R}^d \to \mathbb{R}^M$ takes as input the $d$-dimensional patch embedding and its corresponding GMM responsibilities, and computes an $M$-dimensional component-wise aggregated representation. In particular, $\phi^i(\cdot,\cdot)$ uses the responsibility vector to weight the contribution of each patch with respect to the Gaussian components, producing a post-aggregation prototype-level embedding for downstream fusion.

In our formulation, each Gaussian component contributes $(2d+1)$ dimensions, consisting of the component mean ($d$ dimensions), the component variance ($d$ dimensions), and the mixture weight (one dimension). Hence, the component-wise representation has size $M = 2d + 1$. With $C$ components in the mixture, the final per-slide representation lies in $\mathbb{R}^{C \times (2d+1)}$.

Given the patch embeddings $\mathbf{Z}^i = \{\mathbf{z}_1^i, \ldots, \mathbf{z}_{N_i}^i\}$ of WSI $i$, the likelihood of the Gaussian mixture model is

$$p(\mathbf{Z}^i; \theta^i) = \prod_{n=1}^{N_i} p(\mathbf{z}_n^i; \theta^i) = \prod_{n=1}^{N_i} \sum_{c=1}^{C} \pi_c^i \mathcal{N}(\mathbf{z}_n^i; \boldsymbol{\mu}_c^i, \Sigma_c^i), \tag{29}$$

where $\theta^i = \{\pi_c^i, \boldsymbol{\mu}_c^i, \Sigma_c^i\}_{c=1}^{C}$ are the parameters of the mixture.

The log-likelihood is

$$\ell(\theta^i) = \sum_{n=1}^{N_i} \log \left( \sum_{c=1}^{C} \pi_c^i \mathcal{N}(\mathbf{z}_n^i; \boldsymbol{\mu}_c^i, \Sigma_c^i) \right). \tag{30}$$

Direct maximization of $\ell(\theta^i)$ is intractable due to the summation inside the logarithm. Instead, we apply the Expectation-Maximization algorithm.

**E-step.** For each observation $\mathbf{z}_n^i$, the posterior probability that it belongs to component $c$ under parameters $\theta^{i(t)}$ is

$$p(c_n^i = c \mid \mathbf{z}_n^i; \theta^{i(t)}) = \frac{\pi_c^{i(t)} \mathcal{N}(\mathbf{z}_n^i; \mu_c^{i(t)}, \Sigma_c^{i(t)})}{\sum_{c'=1}^{C} \pi_{c'}^{i(t)} \mathcal{N}(\mathbf{z}_n^i; \mu_{c'}^{i(t)}, \Sigma_{c'}^{i(t)})}. \tag{31}$$

**M-step.** The parameters are updated by maximizing the expected complete-data log-likelihood:

$$\pi_c^{i(t+1)} = \frac{1}{N_i} \sum_{n=1}^{N_i} p(c_n^i = c \mid \mathbf{z}_n^i; \theta^{i(t)}), \tag{32}$$

$$\boldsymbol{\mu}_c^{i(t+1)} = \frac{\sum_{n=1}^{N_i} p(c_n^i = c \mid \mathbf{z}_n^i; \theta^{i(t)}) \, \mathbf{z}_n^i}{\sum_{n=1}^{N_i} p(c_n^i = c \mid \mathbf{z}_n^i; \theta^{i(t)})}, \tag{33}$$

$$\Sigma_c^{i(t+1)} = \frac{\sum_{n=1}^{N_i} p(c_n^i = c \mid \mathbf{z}_n^i; \theta^{i(t)}) \big(\mathbf{z}_n^i - \boldsymbol{\mu}_c^{i(t+1)}\big)\big(\mathbf{z}_n^i - \boldsymbol{\mu}_c^{i(t+1)}\big)^{\top}}{\sum_{n=1}^{N_i} p(c_n^i = c \mid \mathbf{z}_n^i; \theta^{i(t)})}. \tag{34}$$

In practice, we initialize the mixture weights as uniform ($\pi_c^i = 1/C$), set $\boldsymbol{\mu}_c^i$ using $k$-means centroids, and initialize $\Sigma_c^i$ as diagonal matrices. The EM algorithm is iterated until convergence of $\ell(\theta^i)$.

### A.4 CONSISTENCY BETWEEN MIXTURE OF GRFNS AND GMM

The consistency between Gaussian mixture models (GMM) and mixture Gaussian random fuzzy numbers (m-GRFN) can be clarified by their component structures. In a GMM, each observation is

generated from one of $C$ Gaussian components,

$$p(\mathbf{z}) = \sum_{c=1}^{C} \pi_c \mathcal{N}(\mathbf{z}; \boldsymbol{\mu}_c, \sigma_c^2), \tag{35}$$

where $\pi_c$ denotes the prior weight. Each component describes the feature distribution under a conditional probability. In the evidential framework, the same conditional design is retained, but each Gaussian is replaced with a Gaussian random fuzzy number (GRFN) that enriches the probabilistic description with evidential precision. Hence,

$$\tilde{Y} \sim \sum_{c=1}^{C} \pi_c \tilde{\mathcal{N}}(\mu_c, \sigma_c^2, h_c). \tag{36}$$

If the distributional features of each component can be consistently mapped to evidential quantities, then the applicability of m-GRFN naturally coincides with that of GMM. Both models share the same mixture structure and rely on prior weights for aggregation, ensuring aligned application scenarios.

To be noticed, when the evidential precision tends to infinity ($h_c \to +\infty$), the GRFN degenerates into a standard Gaussian random variable. In this limit, the entire m-GRFN reduces exactly to the original GMM. This shows that GMM can be viewed as a special case of m-GRFN, while m-GRFN itself provides a principled evidential generalization of the classical mixture modeling paradigm.

## B    EXPERIMENTAL DETAILS

### B.1    DATASETS

We evaluate DPsurv on five cancer types from The Cancer Genome Atlas (TCGA), where $n$ denotes the number of patients and WSI denotes the number of whole-slide images: Breast Invasive Carcinoma (BRCA, $n = 1,041$, WSI=1,111), Bladder Urothelial Carcinoma (BLCA, $n = 373$, WSI=437), Uterine Corpus Endometrial Carcinoma (UCEC, $n = 504$, WSI=565), Kidney Renal Clear Cell Carcinoma (KIRC, $n = 511$, WSI=517), and Lung Adenocarcinoma (LUAD, $n = 456$, WSI=1,024). During preprocessing, we removed duplicate cases and excluded samples with missing survival time.

### B.2    BASELINES

For baseline training, we use the AdamW optimizer with a learning rate of $1 \times 10^{-4}$, a weight decay of $1 \times 10^{-5}$, and a cosine learning-rate decay scheduler. Supervised baselines are trained with the negative log-likelihood loss for 20 epochs using a batch size of one patient. For PANTHER, we instead employ the Cox proportional hazards loss, training for 50 epochs with a batch size of 64 patients.

We compare DPsurv with the following representative methods:

- **ABMIL**: An attention-based MIL framework originally designed for WSI classification; in this work, we adapt it for discrete survival prediction.
- **TransMIL**: A Transformer-based MIL model for WSI classification; here, we modify it for discrete survival prediction.
- **DSMIL**: A dual-stream MIL model originally for WSI classification; we adapt it to discrete survival prediction.
- **ILRA**: An instance-level representation aggregation MIL model, developed for classification, and extended here to *discrete survival prediction*.
- **BayesMIL**: A Bayesian MIL framework proposed for WSI classification, adapted in our study for discrete survival prediction.
- **AttnMISL**: A survival-specific MIL model; for consistency with other methods, we implement it in the discrete survival prediction setting.

- **PANTHER**: A prototype-based survival model; we also adapt it to the discrete survival prediction setting for consistent evaluation.

- **EVREG**: An evidential regression model originally proposed for tabular data; in our setting, WSI features are reduced by PCA and clustered via GMM before applying EVREG with its original loss function.

- **UMSA**: A multiscale survival model guided by genomic data; since molecular inputs are unavailable in our setting, we adapt it to a single-scale version using only WSI features.

For all baselines, we unify the survival time discretization into a four-class quantile setting, consistent with traditional survival formulations and enabling direct probability outputs for calibration assessment.

### B.3 EVALUATION CRITERIA

To comprehensively evaluate survival prediction models, we employ three widely used metrics: the concordance index (C-index), the integrated Brier score (IBS), and the integrated binomial log-likelihood (IBLL). Together, these metrics assess both the discriminative ability and calibration quality of survival estimates under censoring.

**Concordance Index.** The C-index quantifies a model's discriminative power by measuring the agreement between predicted risks and actual survival outcomes. It is defined as the proportion of all comparable subject pairs whose predictions are correctly ordered:

$$\text{C-index} = \frac{\sum_{i,j} \mathbf{1}(T_i < T_j)\,\mathbf{1}(\hat{r}_i > \hat{r}_j)}{\sum_{i,j} \mathbf{1}(T_i < T_j)}, \tag{37}$$

where $T_i$ and $T_j$ are observed times, $\hat{r}_i$ is the predicted risk score, and $\mathbf{1}(\cdot)$ is the indicator function. A C-index of $0.5$ indicates random ranking, while values closer to $1$ suggest near-perfect concordance.

**Integrated Brier Score.** The Brier score (BS) measures the squared error between predicted survival probabilities $\hat{S}(t|x_i)$ and observed binary survival outcomes at a fixed time $t$. To account for right censoring, inverse-probability weights are introduced via the Kaplan–Meier estimate $\hat{G}(\cdot)$ of the censoring distribution:

$$\text{BS}(t) = \frac{1}{N} \sum_{i=1}^{N} \left[ \frac{\hat{S}(t|x_i)^2\,\mathbf{1}(T_i \leq t, D_i = 1)}{\hat{G}(T_i)} + \frac{(1 - \hat{S}(t|x_i))^2\,\mathbf{1}(T_i > t)}{\hat{G}(t)} \right]. \tag{38}$$

Aggregating BS over a time interval $[t_1, t_2]$ yields the integrated Brier score:

$$\text{IBS} = \frac{1}{t_2 - t_1} \int_{t_1}^{t_2} \text{BS}(s)\,ds. \tag{39}$$

A lower IBS value indicates more accurate and better calibrated survival probability predictions.

**Integrated Binomial Log-Likelihood.** The binomial log-likelihood (BLL) evaluates the fit of predicted probabilities to observed outcomes and, analogously to the Brier score, incorporates inverse censoring weights via $\hat{G}(\cdot)$:

$$\text{BLL}(t) = \frac{1}{N} \sum_{i=1}^{N} \left[ \frac{\log\big(1 - \hat{S}(t|x_i)\big)\,\mathbf{1}(T_i \leq t, D_i = 1)}{\hat{G}(T_i)} + \frac{\log\big(\hat{S}(t|x_i)\big)\,\mathbf{1}(T_i > t)}{\hat{G}(t)} \right]. \tag{40}$$

The integrated version over $[t_1, t_2]$ is given by

$$\text{IBLL} = \frac{1}{t_2 - t_1} \int_{t_1}^{t_2} \text{BLL}(s)\,ds. \tag{41}$$

## B.4 DPSURV

**Evidential Neural Network Initialization.** We initialize the model by a weighted K-means algorithm. For each Gaussian component, we drop samples with small mixture proportions ($\hat{\pi}_c \leq \tau$), $\ell_2$-normalize the remaining features, and run weighted K-means with $\hat{\pi}_c$ as sample weights. the resulting cluster centroids serve as slide-level prototype vectors $\boldsymbol{p}_{c,k}$. For each slide-level prototype, we aggregate the survival responses of its assigned samples to initialize the evidential parameters: set $\beta_{c,k0}$ to the weighted mean of the log survival time, $\sigma_{c,k}^2$ to the corresponding weighted variance, and initialize $\beta_{c,k} = 0$. The epistemic uncertainty parameter $h_{c,k}$ is set to be 4 and scaled by the average mixture proportion within the cluster, thereby discounting the evidence of slide-level prototypes. The decay parameter $\gamma_{c,k}$ is initialized proportional to the inverse square root of the average squared cosine distance within the cluster.

**Hyperparameter Settings.** For model training, we set the number of patch prototypes to 16 and initialize $k$-means prototypes using 100,000 sampled patches. The AdamW optimizer with a cosine learning rate scheduler is adopted across all experiments. All experiments are conducted on NVIDIA A100 GPUs with 80GB memory. The detailed hyperparameter configurations for different datasets are summarized in Table 3.

Table 3: Hyperparameter settings for different datasets.

| Dataset | Learning rate | Epoch | Batch size | Weight decay | $\xi$ | $\rho$ | $\tau$ | $K$ | $\lambda$ |
|---------|---------------|-------|------------|--------------|-------|--------|--------|-----|-----------|
| BLCA | 2e−4 | 50 | 32 | 2e−4 | 0.01 | 0.01 | 0.01 | 2–4 | 0.1 |
| KIRC | 2e−4 | 50 | 32 | 2e−4 | 0.01 | 0.01 | 0.1 | 4 | 0.1 |
| UCEC | 2e−4 | 50 | 32 | 2e−4 | 0.01 | 0.01 | 0.1 | 5 | 0.1 |
| BRCA | 5e−5 | 50 | 32 | 2e−4 | 0.01 | 0.01 | 0.01 | 2 | 0.1 |
| LUAD | 2e−6 | 50 | 32 | 2e−4 | 0.01 | 0.01 | 0.01 | 2–3 | 0.1 |

**Learning rate.** Several hyperparameters are adapted to the characteristics of different cancer cohorts. The learning rate is tuned according to both dataset scale and optimization stability. For example, LUAD, which exhibits relatively high heterogeneity across slides, requires a smaller learning rate (2e−6) to stabilize training, while more homogeneous cohorts such as BLCA, KIRC, and UCEC can be optimized with a higher rate (2e−4).

**Prototype initialization threshold** $\tau$**.** The parameter $\tau$ is used during prototype initialization to filter out samples whose assignment probability $\pi$ falls below $\tau$. In cancers with low morphological heterogeneity, assignment probabilities are generally higher and more concentrated, which allows the use of a larger $\tau$ to discard low-confidence samples and improve the reliability of evidence associated with each prototype. Conversely, in highly heterogeneous cancers, where assignment probabilities are more dispersed, a smaller $\tau$ is preferred to retain sufficient diversity in the initialization.

**Rationale for the number of prototypes** $K$**.** The choice of $K$ reflects both the potential number of histological phenotypes or subtypes within a cancer type and the effective training size in each fold. In particular, since five-fold cross-validation produces variable training sizes across folds, we sometimes set $K$ as a range (e.g., 2–4 for BLCA, 2–3 for LUAD), so that larger folds with more samples can support a richer set of prototypes while smaller folds avoid over-parameterization.

- **BLCA** ($K = 2$–$4$). Urothelial carcinoma mainly exhibits a few dominant structural patterns, but when the training split is larger, additional prototypes (up to 4) help capture secondary variations.
- **KIRC** ($K = 4$). Clear-cell renal carcinoma slides frequently contain multiple co-existing phenotypes (e.g., stromal variation, necrosis). With sufficient training data, $K = 4$ balances diversity and stability.

- **UCEC** ($K = 5$). Endometrial carcinoma is characterized by substantial morphological diversity, and the dataset size supports a relatively large number of prototypes, making $K = 5$ appropriate.

- **BRCA** ($K = 2$). Although the training set is large, breast carcinoma WSIs are often dominated by a few invasive growth patterns. Thus, a small $K$ suffices to capture the major morphological modes without redundancy.

- **LUAD** ($K = 2$–$3$). Lung adenocarcinoma encompasses several architectural patterns, but uneven fold sizes and skewed subtype distributions suggest using a compact $K$. Larger folds allow up to 3 prototypes, while smaller folds are constrained to 2.

## C    ADDITIONAL RESULT

Table 4 reports the IBLL results from 5-fold cross-validation across five TCGA cohorts. Among methods without uncertainty awareness (UA), PANTHER achieves the best performance with an average IBLL of 1.350, highlighting the benefit of prototype-based reasoning over conventional MIL approaches such as ABMIL, TransMIL, DSMIL, and ILRA. On average, models that incorporate UA achieve lower IBLL compared to those without UA, indicating the effectiveness of explicitly modeling uncertainty for improving calibration. Most notably, our proposed DPsurv further advances this trend and consistently outperforms all baselines, attaining the lowest average IBLL (0.862) across datasets. This demonstrates that DPsurv not only preserves strong discrimination but also provides substantially better calibrated survival predictions.

Table 4: IBLL from 5-fold cross-validation on five TCGA datasets, with the averaged IBLL across datasets (Avg). Best results are in **bold**; second best are underlined.

| | Method | BRCA | BLCA | LUAD | UCEC | KIRC | Avg ($\downarrow$) |
|---|---|---|---|---|---|---|---|
| ✗ UA | ABMIL | 3.924 (±1.52) | 2.394 (±0.56) | 2.712 (±1.14) | 4.377 (±1.39) | 2.706 (±1.24) | 3.223 |
| | TransMIL | 8.265 (±2.08) | 5.143 (±0.87) | 4.878 (±1.18) | 6.092 (±0.74) | 4.689 (±1.55) | 5.813 |
| | DSMIL | 2.718 (±0.99) | 1.231 (±0.27) | 1.396 (±0.33) | 2.251 (±0.86) | 1.587 (±0.67) | 1.836 |
| | AttnMISL | 6.481 (±1.21) | 2.569 (±0.95) | 3.047 (±1.17) | 5.685 (±1.44) | 3.199 (±1.34) | 4.196 |
| | ILRA | 8.677 (±0.64) | 5.641 (±0.34) | 5.682 (±1.74) | 6.236 (±1.81) | 5.570 (±1.62) | 6.361 |
| | PANTHER | 1.834 (±0.33) | 0.952 (±0.08) | **1.044** (±0.05) | 1.757 (±0.11) | 1.163 (±0.27) | 1.350 |
| ✓ UA | EVREG | 4.705 (±1.99) | 4.643 (±1.56) | 4.780 (±1.08) | 4.691 (±1.67) | 4.192 (±0.95) | 4.602 |
| | UMSA | 4.321 (±2.15) | 2.195 (±0.62) | 2.785 (±1.13) | 4.037 (±1.09) | 2.668 (±1.04) | 3.201 |
| | BayesMIL | 2.862 (±1.02) | 1.462 (±0.28) | 1.925 (±0.52) | 3.066 (±0.58) | 1.833 (±0.78) | 2.230 |
| | DPsurv (ours) | **0.539** (±0.09) | **0.857** (±0.35) | 1.501 (±0.15) | **0.676** (±0.16) | **0.737** (±0.12) | **0.862** |

## D    LIMITATION AND DISCUSSION

One limitation of our framework lies in the sensitivity to the number of prototypes used at different stages of the modeling pipeline. During GMM clustering, the number of patch prototypes substantially influences the representational quality. For cancers with high morphological heterogeneity, using a larger number of patch prototypes is desirable to capture diverse histological phenotypes, whereas for cancers with relatively low heterogeneity, fewer prototypes are sufficient to retain meaningful morphological semantics. Similarly, the number of component prototypes is affected by both dataset size and histological diversity. Excessively many or too few component prototypes may reduce interpretability, either by introducing redundant clusters or by oversimplifying heterogeneous patterns. Therefore, careful selection of prototype numbers remains critical for ensuring robust performance and interpretability.

Our proposed DPsurv framework not only provides accurate survival prediction but also offers interpretability through prototype-based evidence modeling. By identifying representative patch- and component-level prototypes, the model can uncover potential histological phenotypes that correspond to typical morphological patterns. Such discoveries may aid pathologists in establishing new

classification rules for differentiating subtypes within the same tissue type, thereby facilitating more refined diagnostic systems. Furthermore, because our model quantifies the risk associated with each morphological phenotype, it has the potential to reveal hidden prognostic factors. This ability to link specific tissue patterns with survival risk highlights the clinical utility of DPsurv, suggesting that it may serve as a valuable tool for both precision prognosis and exploratory pathology research.

# E    USE OF LARGE LANGUAGE MODELS

We used a Large Language Model (LLM), specifically ChatGPT (GPT-5), to assist with improving the presentation of this manuscript. The LLM was employed for language polishing, grammar checking, and LaTeX formatting suggestions. The conceptual design of the study, algorithm development, experimental implementation, and result analysis were entirely conducted by the authors without LLM assistance.

# F    ABLATION STUDY

We decomposed the Dual-Prototype Architecture into four ablation parts: GMM with patch prototypes, GRFN for uncertainty quantification, Component prototype for evidnece modeling and Evidence Mixture. The results in Table 5 show a clear gain of each module: compared with using only GMM, GRFN substantially improves prediction calibration, component prototypes and evidence mixture provide additional gains. The full model achieves the highest discriminative performance without compromising calibration quality.

Table 5: Ablation study on the DPsurv architecture across five TCGA datasets. Metrics are reported as mean values. **GRFN** indicates whether the evidential GRFN architecture is enabled; **Component Prototype** denotes whether component-level prototype aggregation is used; **Evidence Mixture** refers to whether evidence mixture fusion is applied.

| GMM | GRFN | Component Prototype | Evidence Mixture | C-index | IBS | IBLL |
|:---:|:---:|:---:|:---:|:---:|:---:|:---:|
| ✓ | ✗ | ✗ | ✗ | 0.6584 | 0.5851 | 1.5743 |
| ✓ | ✓ | ✗ | ✗ | 0.6634 | 0.3305 | 0.8937 |
| ✓ | ✓ | ✓ | ✗ | 0.5380 | 0.3003 | 0.9529 |
| ✓ | ✓ | ✗ | ✓ | 0.6445 | **0.2197** | **0.7203** |
| ✓ | ✓ | ✓ | ✓ | **0.6850** | 0.2880 | 0.8620 |

# G    HYPERPARAMETER STUDY

We conduct a hyperparameter sensitivity analysis to study the robustness of our proposal with those parameters on two cancer datasets, KIRC and UCEC. The results shown in Tables 6–7 indicate that performance varies slightly across a broad range of these settings in 5-fold cross-site validation, suggesting that DPsurv is robust and does not require heavy hyperparameter tuning. For example, the C-index for KIRC remains within a relatively tight band of approximately 0.72–0.74 across many choices of $K$, $C$, and $\lambda$, and UCEC similarly stays around 0.70–0.72.

# H    COMPUTATIONAL COST

We evaluate the computation cost of our model compared with simpler MIL models (The results are shown in Table 8). To provide a representative comparison, we report results on BLCA (437 WSIs), the smallest cohort, and BRCA (1111 WSIs), the largest cohort in our study. Both datasets contain very high–resolution WSIs, enabling us to assess computational behavior under both low- and high-volume settings.

As shown in Table 8, the overall computational overhead of DPsurv remains relatively small. This efficiency is largely attributed to the GMM aggregation, which compresses each WSI into a compact

Table 6: Hyperparameter sensitivity analysis on KIRC dataset. Metrics are reported as mean $\pm$ standard deviation.

| Parameter | Setting | C-index | IBS | NBLL |
|---|---|---|---|---|
| $K$ | 2 | 0.7271±0.0933 | 0.2068±0.0257 | 0.6049±0.0570 |
| | 3 | 0.7289±0.0818 | 0.2219±0.0427 | 0.6388±0.1019 |
| | 4 | 0.7386±0.1029 | 0.2574±0.0484 | 0.7368±0.1210 |
| | 5 | 0.7166±0.0973 | 0.2800±0.0467 | 0.8013±0.1168 |
| | 6 | 0.7217±0.0833 | 0.2925±0.0437 | 0.8436±0.1086 |
| $C$ | 8 | 0.7287±0.0671 | 0.2759±0.0517 | 0.8136±0.1652 |
| | 16 | 0.7386±0.1029 | 0.2574±0.0484 | 0.7368±0.1210 |
| | 32 | 0.7176±0.0881 | 0.2375±0.0548 | 0.6838±0.1403 |
| $\lambda$ | 0.1 | 0.7386±0.1029 | 0.2574±0.0484 | 0.7368±0.1210 |
| | 0.3 | 0.7306±0.0936 | 0.2078±0.0287 | 0.5947±0.0605 |
| | 0.5 | 0.7261±0.0953 | 0.1788±0.0250 | 0.5262±0.0580 |
| | 0.7 | 0.7268±0.1031 | 0.1615±0.0362 | 0.4885±0.1037 |
| | 0.9 | 0.7267±0.1030 | 0.1556±0.0518 | 0.4828±0.1445 |
| $\alpha$ | 0.1 | 0.7300±0.0894 | 0.2202±0.0365 | 0.6337±0.0878 |
| | 0.3 | 0.7295±0.0914 | 0.2316±0.0460 | 0.6621±0.1037 |
| | 0.5 | 0.7386±0.1029 | 0.2574±0.0484 | 0.7368±0.1210 |
| | 0.7 | 0.7267±0.0986 | 0.2918±0.0547 | 0.8381±0.1727 |
| | 0.9 | 0.7120±0.1010 | 0.4079±0.0458 | 1.2030±0.1472 |
| $R_1$ | 0 | 0.7310±0.0931 | 0.2479±0.0417 | 0.7013±0.0931 |
| | 0.005 | 0.7273±0.0948 | 0.2520±0.0416 | 0.7131±0.0940 |
| | 0.01 | 0.7386±0.1029 | 0.2574±0.0484 | 0.7368±0.1210 |
| | 0.1 | 0.7309±0.0930 | 0.2502±0.0431 | 0.7096±0.1019 |
| $R_2$ | 0 | 0.7291±0.1019 | 0.2450±0.0387 | 0.6973±0.0954 |
| | 0.005 | 0.7287±0.1023 | 0.2513±0.0399 | 0.7125±0.0964 |
| | 0.01 | 0.7386±0.1029 | 0.2574±0.0484 | 0.7368±0.1210 |
| | 0.1 | 0.7333±0.0905 | 0.2473±0.0399 | 0.7015±0.0911 |

Table 7: Hyperparameter sensitivity analysis on UCEC dataset. Metrics are reported as mean $\pm$ standard deviation.

| Parameter | Setting | C-index | IBS | NBLL |
|---|---|---|---|---|
| $K$ | 2 | 0.7491±0.0948 | 0.1366±0.0498 | 0.4257±0.1152 |
| | 3 | 0.7190±0.0870 | 0.2440±0.0589 | 0.6764±0.1576 |
| | 4 | 0.7106±0.1156 | 0.2158±0.0571 | 0.6098±0.1440 |
| | 5 | 0.7063±0.1023 | 0.2271±0.0473 | 0.6354±0.1222 |
| | 6 | 0.7309±0.0822 | 0.2398±0.0560 | 0.6727±0.1598 |
| $C$ | 8 | 0.7293±0.0719 | 0.2698±0.0760 | 0.7545±0.2052 |
| | 16 | 0.7190±0.0870 | 0.2440±0.0589 | 0.6764±0.1576 |
| | 32 | 0.7185±0.0881 | 0.2375±0.0548 | 0.6838±0.1403 |
| $\lambda$ | 0.1 | 0.7190±0.0870 | 0.2440±0.0589 | 0.6764±0.1576 |
| | 0.3 | 0.7097±0.0889 | 0.1778±0.0420 | 0.5185±0.1114 |
| | 0.5 | 0.7099±0.0852 | 0.1440±0.0427 | 0.4432±0.1172 |
| | 0.7 | 0.7104±0.0879 | 0.1203±0.0455 | 0.3916±0.1280 |
| | 0.9 | 0.7108±0.0859 | 0.1113±0.0485 | 0.3718±0.1410 |
| $\alpha$ | 0.1 | 0.6927±0.0935 | 0.1839±0.0542 | 0.5348±0.1486 |
| | 0.3 | 0.7022±0.0967 | 0.2077±0.0507 | 0.5882±0.1345 |
| | 0.5 | 0.7190±0.0870 | 0.2440±0.0589 | 0.6764±0.1576 |
| | 0.7 | 0.7093±0.0950 | 0.2886±0.0806 | 0.8263±0.2608 |
| | 0.9 | 0.7140±0.0921 | 0.3840±0.0644 | 1.1160±0.2484 |
| $R_1$ | 0 | 0.7076±0.0943 | 0.2287±0.0473 | 0.6362±0.1247 |
| | 0.005 | 0.7072±0.0997 | 0.2284±0.0511 | 0.6351±0.1310 |
| | 0.01 | 0.7190±0.0870 | 0.2440±0.0589 | 0.6764±0.1576 |
| | 0.1 | 0.7049±0.1050 | 0.2251±0.0527 | 0.6293±0.1383 |
| $R_2$ | 0 | 0.7062±0.0947 | 0.2261±0.0471 | 0.6317±0.1243 |
| | 0.005 | 0.7066±0.0948 | 0.2270±0.0474 | 0.6346±0.1242 |
| | 0.01 | 0.7190±0.0870 | 0.2440±0.0589 | 0.6764±0.1576 |
| | 0.1 | 0.7072±0.0931 | 0.2276±0.0469 | 0.6353±0.1241 |

set of mixture features and keeps the downstream evidential model lightweight. Moreover, the similar runtime pattern observed for BLCA and BRCA suggests that DPsurv scales well to larger cohorts and can accommodate higher-resolution WSIs.

Table 8: Training time comparison (in seconds) across methods on BLCA and BRCA datasets.

| Method | BLCA | BRCA |
|---|---|---|
| ABMIL | 956.91 | 1747.28 |
| TransMIL | 1449.01 | 3109.04 |
| BayesMIL | 959.23 | 2224.95 |
| Panther | 291.78 | 381.22 |
| DPsurv | 550.86 | 972.25 |

# I EXTENDED INTERPRETABILITY ANALYSIS

## I.1 ADDTIONAL EXAMPLES

To reduce concerns of cherry-picking, we include two additional randomly selected whole-slide examples in Figure 4 and Figure 5. Each example presents the assignment map and relative risk visualization.

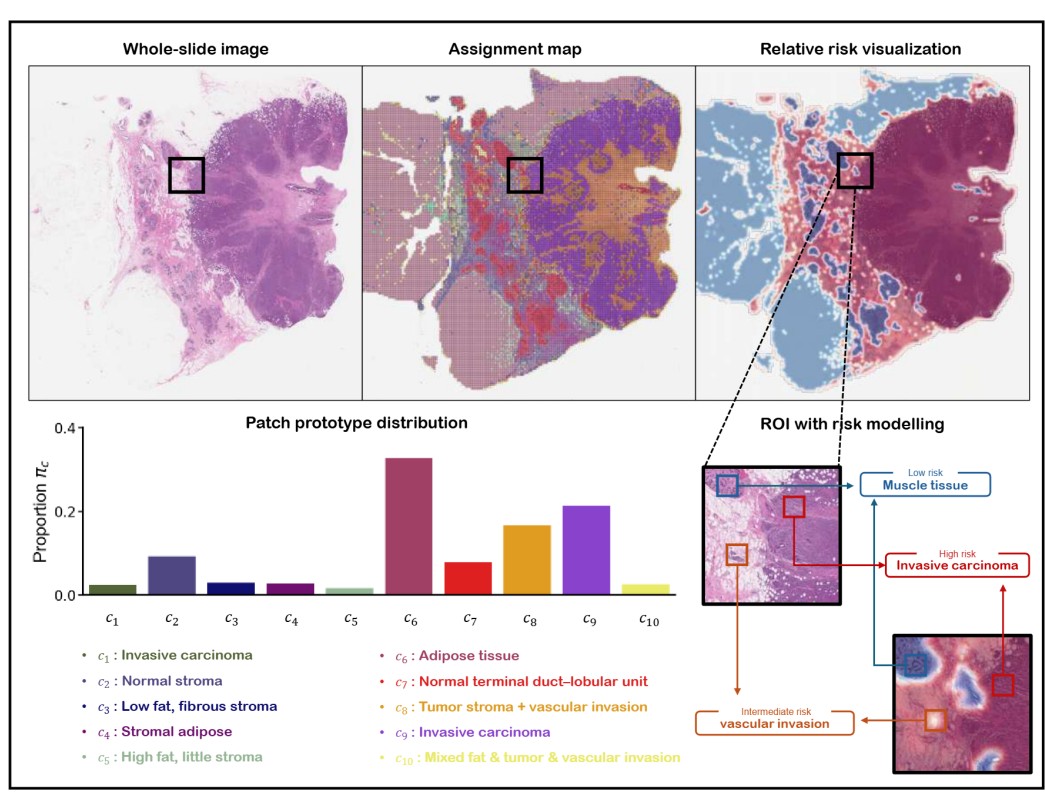

Figure 4: Additional interpretation of the DPsurv in WSI survival prediction.

## I.2 OBJECTIVE EVALUATION

We additionally performed a small-scale clinical coherence evaluation. A subset of slides was randomly sampled, and for each case we generated both the GMM-based assignment map and the

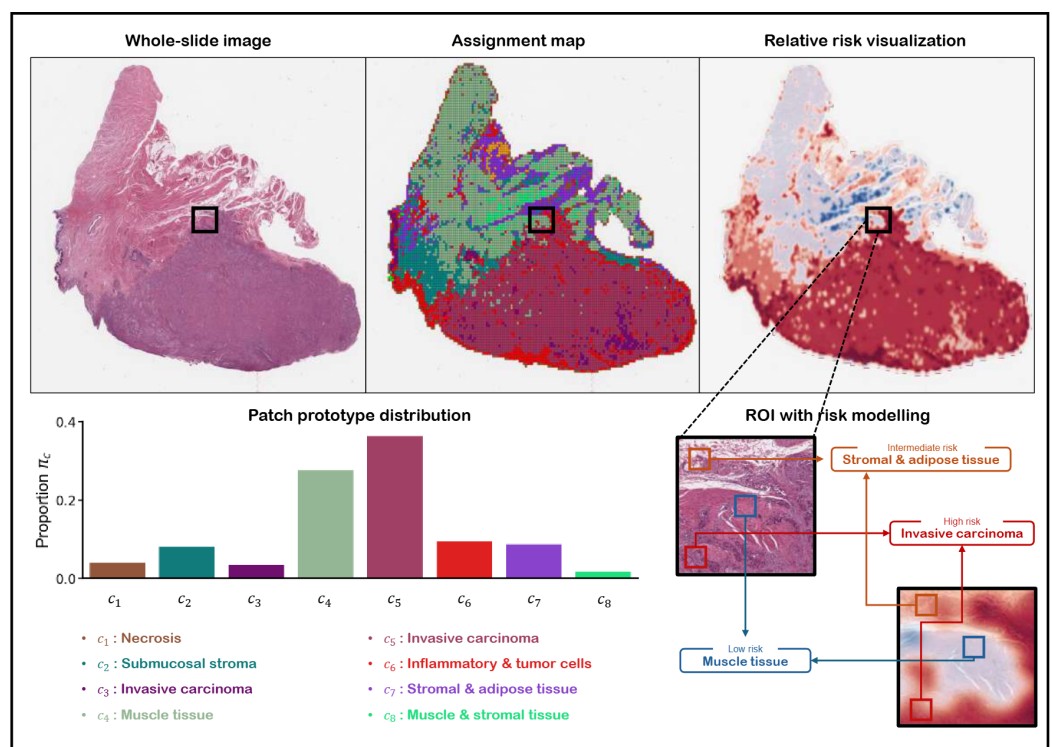

Figure 5: Additional interpretation of the DPsurv in WSI survival prediction.

corresponding relative risk visualization produced by DPsurv. Two board-certified pathologists independently assessed the coherence between the model-derived interpretations and their own clinical judgement, using a 10-point ordinal scale.

As summarized in Table 9, the coherence scores are consistently high across cases. Furthermore, the minimal difference between the average ratings for the assignment map and the risk visualization indicates that when the GMM-derived component assignments align well with pathologists' expectations, the downstream risk estimation produced by DPsurv is also more likely to be clinically reliable. This provides additional evidence supporting the interpretability and clinical plausibility of our model's reasoning process.

Table 9: Coherence between model-generated interpretations and clinical insights.

| Case ID | Relative Risk Visualization | Assignment Map |
|---|---|---|
| TCGA-4Z-AA7S | 7 | 8 |
| TCGA-93-8067 | 9 | 9 |
| TCGA-A2-A3XW | 8 | 9 |
| TCGA-UY-A78L | 7 | 9 |
| TCGA-XF-A8HB | 7 | 8 |
| TCGA-XF-A9SH | 7 | 8 |
| TCGA-XF-A9ST | 8 | 9 |
| TCGA-XF-AAMJ | 4 | 8 |
| TCGA-XF-AAMZ | 7 | 7 |
| TCGA-ZF-AA4W | 6 | 7 |
| **Overall Average** | **7.0** | **8.2** |

## J    TOY EXAMPLE

Consider two component prototypes associated with component $c$,

$$\tilde{Y}_{c,1} \sim \tilde{N}(\mu_{c,1}, \sigma^2_{c,1}, h_{c,1}), \qquad \tilde{Y}_{c,2} \sim \tilde{N}(\mu_{c,2}, \sigma^2_{c,2}, h_{c,2}),$$

together with their similarity scores $s_{c,1}, s_{c,2} \in [0,1]$. Using the fusion rule , the fused GRFN is

$$\tilde{Y}_c \sim \tilde{N}(\mu_c, \sigma^2_c, h_c),$$

with parameters

$$h_c = s_{c,1} h_{c,1} + s_{c,2} h_{c,2}, \tag{42}$$

$$\mu_c = \frac{s_{c,1} h_{c,1} \mu_{c,1} + s_{c,2} h_{c,2} \mu_{c,2}}{s_{c,1} h_{c,1} + s_{c,2} h_{c,2}}, \tag{43}$$

$$\sigma^2_c = \frac{s^2_{c,1} h^2_{c,1} \sigma^2_{c,1} + s^2_{c,2} h^2_{c,2} \sigma^2_{c,2}}{(s_{c,1} h_{c,1} + s_{c,2} h_{c,2})^2}. \tag{44}$$

Foe example, Let

$$\mu_{c,1} = 2, \ \mu_{c,2} = 7, \qquad \sigma^2_{c,1} = 1, \ \sigma^2_{c,2} = 4,$$
$$h_{c,1} = 1.2, \ h_{c,2} = 0.8, \qquad s_{c,1} = 0.6, \ s_{c,2} = 0.3.$$

Substituting into equation 42–equation 44 yields

$$h_c = 0.96, \qquad \mu_c = 3.25, \qquad \sigma^2_c \approx 0.812.$$

Thus, the fused GRFN is

$$\tilde{Y}_c = \tilde{N}(3.25, \ 0.812, \ 0.96).$$

## K    COMPARISON WITH ALTERNATIVE UNCERTAINTY QUANTIFICATION APPROACH

We conducte extra experimental comparisons of our method and two strong non-evidential uncertainty baselines, Monte-Carlo (MC) dropout and Deep ensemble. The comparison results in Table 10 show superior predictive performance across all three evaluation metrics compared with the MC dropout and Deep ensemble. To make it clearer, we highlight the key differences between GRFN-based uncertainty modeling and traditional non-evidential baselines here:

**(1) Disentangling epistemic and aleatoric uncertainty.** MC Dropout and Deep Ensemble estimate uncertainty via parameter sampling or model ensembling, but they do not distinguish between epistemic uncertainty and aleatoric uncertainty. In contrast, GRFN $\tilde{Y} \sim \tilde{N}(\mu, \sigma^2, h)$ explicitly models both uncertianties at the evidence level, i.e., $\sigma$ and $h \in [0, +\infty)$ represent the aleatoric and epistemic uncertainties.

**(2) Predictive calibration through uncertainty weighting.** While MC Dropout and Deep Ensemble only quantify uncertainty, GRFN not only estimates uncertainty but also leverages it to optimize model behavior. Specifically, evidences with higher uncertainty are automatically assigned lower weights during aggregation, enabling GRFN to make more robust and interpretable predictions.

Table 10: Comparison of uncertainty modeling approaches on survival prediction.

| Method | C-index | IBS | IBLL |
|---|---|---|---|
| MC Dropout | 0.6454 | 0.7743 | 2.8195 |
| Deep Ensemble | 0.6483 | 0.8033 | 3.5054 |
| **Ours** | **0.6850** | **0.2880** | **0.8620** |

In addition, we provide calibration plots comparing DPsurv with two widely used uncertainty estimation baselines, MC Dropout and Deep Ensemble. As shown in Fig. 6, we report the proportion of $\alpha$-level BPIs and PPIs that contain the uncensored survival times for $\alpha \in 0.1, \ldots, 0.9$ across the five TCGA cohorts (BRCA, BLCA, LUAD, UCEC, and KIRC). These results further demonstrate that DPsurv achieves consistently improved calibration performance relative to both baselines.

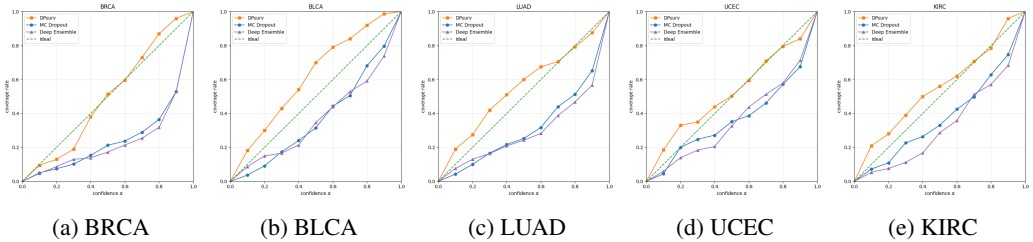

(a) BRCA      (b) BLCA      (c) LUAD      (d) UCEC      (e) KIRC

Figure 6: Calibration plots comparing DPsurv with MC Dropout and Deep Ensemble across five TCGA datasets. The plots show the empirical coverage of $\alpha$-level BPIs and PPIs with respect to uncensored survival times for $\alpha \in 0.1, \ldots, 0.9$. Curves closer to the diagonal indicate better calibration performance.

