# OpenReview forum: "DPsurv: Dual-Prototype Evidential Fusion for Uncertainty-Aware and Interpretable Whole Slide Image Survival Prediction"
_ICLR.cc/2026/Conference — Submitted to ICLR 2026_

### Official Review · Reviewer_Uv5K · 2025-10-19

**Soundness:** 2
**Presentation:** 4
**Contribution:** 2
**Rating:** 4
**Confidence:** 4

**Summary:**

This paper proposes DPsurv, a novel framework for Whole Slide Image (WSI) survival prediction that aims to be both interpretable and uncertainty-aware. The method employs a "dual-prototype" architecture. First, it uses a Gaussian Mixture Model (GMM) on patch embeddings to learn "patch prototypes" that represent distinct morphological phenotypes, yielding a slide-level component embedding. Second, these component embeddings are mapped into an evidence space using "component prototypes" and Gaussian Random Fuzzy Numbers (GRFNs) to model aleatoric and epistemic uncertainty. The final prediction is a mixture of component-level evidence, resulting in a survival curve with uncertainty bounds. The authors claim the framework provides multi-level interpretability (feature, reasoning, and decision) and demonstrate state-of-the-art performance on five TCGA datasets in terms of both discrimination (C-index) and calibration (IBS, IBLL).

**Strengths:**

$\textbf{Significance of the Problem: }$ The work addresses two of the most critical challenges in clinical deep learning for computational pathology: model interpretability and uncertainty quantification. A reliable solution in this space would have a significant clinical impact.

$\textbf{Novelty and Comprehensive Framework: }$ The proposed dual-prototype architecture that models feature-phenotypes and survival evidence separately is a novel concept. The framework provides a comprehensive, end-to-end solution for multi-level interpretability, from patch-level features to component-wise risk contributions, which is a major step beyond simple attention heatmaps.

$\textbf{Excellent Empirical Performance: }$ The method demonstrates state-of-the-art performance across five public TCGA cohorts, achieving the best average C-index for discrimination and, most notably, the best average IBS and IBLL for calibration.  The substantial improvement in calibration metrics is a particularly strong result.

$\textbf{Strong Qualitative Interpretability: }$ The visualizations in Figure 2, supported by pathologist review, provide compelling qualitative evidence that the learned prototypes correspond to meaningful histopathological patterns and that the model's reasoning aligns with clinical knowledge.

**Weaknesses:**

$\textbf{Extreme Complexity and Hyperparameter Sensitivity: }$ The proposed framework is exceptionally complex, combining a foundation model, GMMs, two levels of prototypes, and the evidential framework of GRFNs. The authors themselves acknowledge in the limitations section that the model's performance is sensitive to the number of patch and component prototypes. The hyperparameter table confirms that the number of component prototypes (K) was manually tuned for each dataset, which undermines the robustness and generalizability of the approach.

$\textbf{Crucial Lack of Ablation Studies: }$ Given the model's complexity, the absence of ablation studies is a major omission. It is impossible to assess the contribution of each component to the final performance. For example, how crucial is the second "component prototype" level? What is the performance if the GMM components are mapped directly to survival evidence? How does the complex GRFN framework compare to simpler uncertainty quantification techniques like Deep Ensembles or Monte Carlo Dropout applied to a similar backbone? Without these experiments, the necessity of the full, complex architecture is not justified.

$\textbf{Insufficient Motivation for Evidential Framework Choice: }$ The paper uses Gaussian Random Fuzzy Numbers (GRFNs), a sophisticated but relatively niche framework. While it is shown to work well, the paper does not sufficiently motivate why this specific choice is superior to more widely-used uncertainty quantification methods in deep learning. A discussion comparing its theoretical advantages and disadvantages for this specific problem would strengthen the paper.

**Questions:**

$\textbf{1.}$ Regarding the model's sensitivity to the number of prototypes: This appears to be a major barrier to practical application. Have you considered methods to automatically learn or dynamically determine the optimal number of patch prototypes (C) and component prototypes (K) for a given dataset, rather than relying on manual tuning?

$\textbf{2.}$ Could you please provide ablation studies to justify the complexity of the DPsurv architecture? Specifically, I am interested in understanding the performance impact of (a) removing the second level of component prototypes and mapping GMM outputs directly to survival evidence, and (b) replacing the GRFN-based evidence module with a more standard uncertainty quantification technique like Deep Ensembles to see if similar calibration gains can be achieved with a simpler method.

$\textbf{3.}$ Could you provide more details on the computational cost and scalability of the method, particularly the GMM fitting and the weighted K-means initialization steps? How does the training time scale as the number of WSIs in the training cohort increases?

---

> ### Author Response · Authors · 2025-11-21
> **Official Response to Reviewer Uv5K(1/n)**
>
> **W1 & Q1:** Extreme complexity and hyperparameter sensitivity.
>
> **AW1 & AQ1:** Thanks for your comments and suggestions on the selection of patch prototypes number $C$ and component prototypes number $K$. In our experiments, we performed extensive sensitivity studies covering a wide range of values for both $C$ and $K$, as well as all major trade-off hyperparameters (Table 14 and Table 15). The results show that performance varies only modestly across different settings, with the KIRC C-index consistently remaining around 0.72 to 0.74 and the UCEC C-index around 0.70 to 0.72 across many choices of $K$, $C$, and $\lambda$, indicating that DPsurv is relatively robust and does not rely on precise manual tuning of prototype counts.
>
> We agree that automatically determining the optimal number of prototypes would further improve practicality. In future work, we plan to explore adaptive mechanisms to allow the model to dynamically adjust $C$ and $K$ based on data complexity. We will include this discussion in the revised manuscript.
>
> We have included the results in Appendix G of the revised paper.
>
> ---
>
> Table 14. Hyperparameter study on the KIRC dataset. Metrics are reported as mean values.
>
> | Parameter | Setting | C-index | IBS | IBLL |
> |----------|---------|---------|---------|---------|
> | **K** | 2 | 0.7271 | **0.2068** | **0.6049** |
> |        | 3 | 0.7289 | 0.2219 | 0.6388 |
> |        | 4 | **0.7386** | 0.2574 | 0.7368 |
> |        | 5 | 0.7166 | 0.2800 | 0.8013 |
> |        | 6 | 0.7217 | 0.2925 | 0.8436 |
> | **C** | 8  | 0.7287 | 0.2759 | 0.8136 |
> |        | 16 | **0.7386** | 0.2574 | 0.7368 |
> |        | 32 | 0.7176 | **0.2375** | **0.6838** |
> | **λ** | 0.1 | **0.7386** | 0.2574 | 0.7368 |
> |        | 0.3 | 0.7306 | 0.2078 | 0.5947 |
> |        | 0.5 | 0.7261 | 0.1788 | 0.5262 |
> |        | 0.7 | 0.7268 | 0.1615 | 0.4885 |
> |        | 0.9 | 0.7267 | **0.1556** | **0.4828** |
> | **α** | 0.1 | 0.7300 | **0.2202** | **0.6337** |
> |        | 0.3 | 0.7295 | 0.2316 | 0.6621 |
> |        | 0.5 | **0.7386** | 0.2574 | 0.7368 |
> |        | 0.7 | 0.7267 | 0.2918 | 0.8381 |
> |        | 0.9 | 0.7120 | 0.4079 | 1.2030 |
> | **R₁** | 0     | 0.7310 | **0.2479** | **0.7013** |
> |         | 0.005 | 0.7273 | 0.2520 | 0.7131 |
> |         | 0.01  | **0.7386** | 0.2574 | 0.7368 |
> |         | 0.1   | 0.7309 | 0.2502 | 0.7096 |
> | **R₂** | 0     | 0.7291 | **0.2450** | **0.6973** |
> |         | 0.005 | 0.7287 | 0.2513 | 0.7125 |
> |         | 0.01  | **0.7386** | 0.2574 | 0.7368 |
> |         | 0.1   | 0.7333 | 0.2473 | 0.7015 |
>
> ---
>
> Table 15. Hyperparameter study on the UCEC dataset. Metrics are reported as mean values.
>
> | Parameter | Setting | C-index | IBS | IBLL |
> |----------|---------|---------|---------|---------|
> | **K** | 2 | **0.7491** | **0.1366** | **0.4257** |
> |        | 3 | 0.7190 | 0.2440 | 0.6764 |
> |        | 4 | 0.7106 | 0.2158 | 0.6098 |
> |        | 5 | 0.7063 | 0.2271 | 0.6354 |
> |        | 6 | 0.7309 | 0.2398 | 0.6727 |
> | **C** | 8  | **0.7293** | 0.2698 | 0.7545 |
> |        | 16 | 0.7190 | 0.2440 | 0.6764 |
> |        | 32 | 0.7185 | **0.2375** | **0.6838** |
> | **λ** | 0.1 | **0.7190** | 0.2440 | 0.6764 |
> |        | 0.3 | 0.7097 | 0.1778 | 0.5185 |
> |        | 0.5 | 0.7099 | 0.1440 | 0.4432 |
> |        | 0.7 | 0.7104 | 0.1203 | 0.3916 |
> |        | 0.9 | 0.7108 | **0.1113** | **0.3718** |
> | **α** | 0.1 | 0.6927 | **0.1839** | **0.5348** |
> |        | 0.3 | 0.7022 | 0.2077 | 0.5882 |
> |        | 0.5 | **0.7190** | 0.2440 | 0.6764 |
> |        | 0.7 | 0.7093 | 0.2886 | 0.8263 |
> |        | 0.9 | 0.7140 | 0.3840 | 1.1160 |
> | **R₁** | 0     | 0.7076 | **0.2287** | **0.6362** |
> |         | 0.005 | 0.7072 | 0.2284 | 0.6351 |
> |         | 0.01  | **0.7190** | 0.2440 | 0.6764 |
> |         | 0.1   | 0.7049 | 0.2251 | 0.6293 |
> | **R₂** | 0     | 0.7062 | **0.2261** | **0.6317** |
> |         | 0.005 | 0.7066 | 0.2270 | 0.6346 |
> |         | 0.01  | **0.7190** | 0.2440 | 0.6764 |
> |         | 0.1   | 0.7072 | 0.2276 | 0.6353 |

---

> ### Author Response · Authors · 2025-11-21
> **Official Response to Reviewer Uv5K(2/n)**
>
> **W2 & Q2:** Crucial lack of ablation studies.
>
> **AW2 & AQ2:** Thank you for the suggestions. We provide ablation studies that directly address both aspects raised in the question.
>
> **(a) Removing the second-level component prototypes.**
> This corresponds to the rows in Table 16 where the “CP’’ column is disabled. After removing the second-level component prototypes, the model’s discriminative ability drops substantially. This suggests that CP provides reliable and phenotype-specific evidence for survival modeling, and without such reliable evidence the model can no longer form stable or accurate risk estimates, leading to degraded performance.
>
> **(b) Replacing the GRFN evidence module with simpler uncertainty-estimation methods.**
> Table 17 compares GRFN with MC Dropout and Deep Ensembles. While these approaches provide uncertainty estimates, their calibration (IBS, IBLL) is substantially worse than GRFN. This suggests that simpler uncertainty techniques cannot achieve the same calibration quality in WSI-based survival prediction tasks.
>
> We have included the results in Appendix F&K of the revised paper.
>
> ---
>
> Table 16. Ablation study on the DPsurv architecture across five TCGA datasets. Metrics are reported as mean values.  **GRFN** indicates whether the evidential GRFN architecture is enabled.  **CP** denotes whether component-level prototype aggregation is used.  **EM** refers to whether evidence mixture fusion is applied.
>
> | **GMM** | **GRFN** | **CP** | **EM** | **C-index** | **IBS** | **IBLL** |
> |--------|----------|-----------------------------|---------------------------|-------------|---------|----------|
> | ✓ | ✗ | ✗ | ✗ | 0.6584 | 0.5851 | 1.5743 |
> | ✓ | ✓ | ✗ | ✗ | 0.6634 | 0.3305 | 0.8937 |
> | ✓ | ✓ | ✓ | ✗ | 0.5380 | 0.3003 | 0.9529 |
> | ✓ | ✓ | ✗ | ✓ | 0.6445 | **0.2197** | **0.7203** |
> | ✓ | ✓ | ✓ | ✓ | **0.6850** | 0.2880 | 0.8620 |
>
> ---
>
> Table 17. Comparison of uncertainty modeling approaches on survival prediction across five TCGA datasets. Metrics are reported as mean values.
>
> | Method         | C-index | IBS     | IBLL    |
> |----------------|---------|---------|---------|
> | MC Dropout     | 0.6454  | 0.7743  | 2.8195  |
> | Deep Ensemble  | 0.6483  | 0.8033  | 3.5054  |
> | **Ours**       | **0.6850** | **0.2880** | **0.8620** |

---

> ### Author Response · Authors · 2025-11-21
> **fficial Response to Reviewer Uv5K(3/n)**
>
> **W3:** Insufficient motivation for evidential framework choice.
>
>  **AW3:** Thank you for the insightful comment. Our choice of an evidential framework is motivated by three key factors: (1) its strong ability to model epistemic and aleatoric uncertainty, particularly through GRFN for continuous outcomes; (2) its principled mechanism for combining multi-level information while accounting for uncertainty; and (3) its natural compatibility with prototype-based reasoning for decision-level interpretability.
>
> These properties address core limitations of existing probabilistic survival models, which typically do not capture uncertainty arising from tumour heterogeneity, label censoring, or modeling imperfections. For WSI-based survival prediction, where such uncertainties are substantial, an evidential approach provides a more reliable and interpretable solution. The fusion capability of evidential framework further enables coherent integration of patch- and component-level evidence.
>
> We will add a clearer discussion in the revised manuscript to highlight these motivations.
>
> ---
>
> **Q3:** Could you provide more details on the computational cost and scalability of the method?
>
>  **AQ3:** Thank you for raising the question on computational cost and scalability. We provide additional details below.
>
> To characterize scalability, we report results on BLCA (437 WSIs), the smallest cohort in our study, and BRCA (1111 WSIs), the largest. Both datasets contain very high–resolution WSIs, allowing us to examine behavior under both low- and high-volume settings.
>
> **Cost of GMM fitting.**
> As shown in Table 18, the per-cohort GMM fitting time is relatively small (136.68s for BLCA and 197.61s for BRCA). The increase is approximately linear with respect to the number of WSIs, which is expected because each slide is processed independently after global prototypes are fixed.
>
> **Cost of weighted K-means initialization.**
> The component-prototype initialization step is lightweight, requiring only 4.81s for BLCA and 11.63s for BRCA (Table 18). Since this step operates on slide-level mixture features rather than raw patches, its computational cost scales nearly linearly with the cohort size and remains negligible compared with WSI feature extraction.
>
> **End-to-end training time.**
> The reported DPsurv runtime includes all steps:
> (1) GMM-based WSI embedding extraction,
> (2) weighted K-means component-prototype initialization,
> (3) evidential fusion model training.
>
> As shown in Table 19, the overall overhead of DPsurv remains moderate even for the larger BRCA cohort (972.25s). The similar scaling trend observed in BLCA and BRCA indicates that the computational cost of DPsurv grows close to linearly with the number of WSIs, largely due to the compression effect of GMM aggregation which keeps the downstream evidential model efficient.
>
> Overall, both GMM fitting and weighted K-means steps incur low absolute cost, and the entire pipeline scales favorably as the training cohort increases.
>
> We have included the results in Appendix H of the revised paper.
>
> ---
>
> Table 18. Computational cost of GMM fitting and weighted K-means initialization on BLCA and BRCA datasets (in seconds)
> | **Module**                      | **BLCA** | **BRCA** |
> | ------------------------------- | -------- | -------- |
> | GMM fitting time                | 136.68   | 197.61   |
> | Weighted K-means initialization | 4.81     | 11.63    |
>
> ---
>
> Table 19. Training time comparison (in seconds) across methods on BLCA and BRCA datasets
> | **Method** | **BLCA** | **BRCA** |
> | ---------- | -------- | -------- |
> | ABMIL      | 956.91   | 1747.28  |
> | TransMIL   | 1449.01  | 3109.04  |
> | BayesMIL   | 959.23   | 2224.95  |
> | Panther    | 291.78   | 381.22   |
> | DPsurv     | 550.86   | 972.25   |
>
> _Thank you once again for your detailed and constructive review. We hope that our clarifications and additional analyses have addressed your concerns. We look forward to any further feedback and are happy to provide additional information if needed._
>
> Best Regards,
>
> The Authors

---

### Official Review · Reviewer_MZaC · 2025-10-21

**Soundness:** 3
**Presentation:** 3
**Contribution:** 2
**Rating:** 4
**Confidence:** 4

**Summary:**

This paper proposes a dual‑prototype, evidential fusion framework to tackle the problem that WSI-based survival analysis suffers from limited interpretability and lack of calibrated uncertainty. Specifically, for each slide component, K component prototypes are learned, and evidence is modeled with GRFNs and a mixture of GRFNs is used for the slide level. The experiments showed that the proposed method achieves SOTA and good calibration.

**Strengths:**

1. Problem targeting is timely: uncertainty quantification and interpretable reasoning for WSI survival is both relevant and challenging.
2. Empirical scope: five TCGA cancers with site‑stratified 5‑fold CV is stronger than many prior works.
3. Reported calibration: the comparison between BPIs and PPIs is conceptually apt, and the finding that modeling epistemic uncertainty improves coverage is plausible and clinically valuable.

**Weaknesses:**

1.	Unclear component definition: It is unclear how “component c” in slide i is aligned with “component c” in slide j in order to share $p_{c,k}$ and learn $β_{c,k}$. The text mentions “patch prototypes h_c” and then a GMM, but it is not explicit whether the per‑slide GMM components are anchored to these global patch prototypes.
2.	Choice of $λ$: The paper turns interval‑valued survival bounds (Bel, Pl) into a single S(t) via $S = λ Bel + (1−λ) Pl$, with $λ$ fixed to 0.1. While practical for metrics requiring a point probability, the statistical meaning of this convex interpolation is unclear and may bias calibration if $λ$ is tuned on validation. It also conflates epistemic with a fixed risk tolerance rather than propagating it.
3.	Computational complexity and scalability: Per‑slide GMM in high‑N patch regimes can be expensive; the paper does not detail computational cost vs. alternatives.
4.	Lack of experiments on hyperparameter studies.
5.	Minor Issues: Eq. (3) vs Eq. (4) leaves $φ_i(·,·)$ under‑specified, please write $φ_i$ explicitly. Clarify the final dimensionality $R^{C×(2d+1)}$.

**Questions:**

See Weaknesses.

---

> ### Author Response · Authors · 2025-11-21
> **Official Response to Reviewer MZaC(1/n)**
>
> **W1:** Unclear component definition.
>
> **AW1:** Thank you for highlighting the unclear explanation of the component. We will clarify this section in the final version to avoid potential confusion.  For patch prototypes, we construct them by performing global K-means clustering on millions of patch features extracted from all training-set WSIs. The resulting global prototypes serve as shared anchors across slides and define a common phenotype space for modeling. By fitting the per-slide GMM in this shared prototype space, the component c in different slides is naturally aligned and corresponds to the same underlying phenotype.
>
> ---
>
> **W2:** Choice of $\lambda$.
>
> **AW2:** Thank you for raising the questions about using a fixed $\lambda$. Our decision to set $\lambda = 0.1$ is based on two considerations.
>
> First, convex interpolation between $\mathrm{Bel}(t)$ and $\mathrm{Pl}(t)$ provides a simple and intuitive way to obtain a point-valued survival function when required by standard evaluation metrics. As shown in the Table 9 and Table 10 below, the model performance is relatively insensitive to the choice of $\lambda$, indicating that $\lambda$ does not critically affect calibration or discrimination.
>
> Second, we choose $\lambda = 0.1$ to incorporate information from both $\mathrm{Bel}$ and $\mathrm{Pl}$ while producing a conservative survival estimate [1]. In survival prediction, underestimating risk can be clinically undesirable, and weighting the lower bound slightly more aligns with this safety consideration.
>
> We will clarify this choice in the revised manuscript and discuss it as a future direction for improving our model.
>
> [1] Denoeux, Thierry. "Quantifying prediction uncertainty in regression using random fuzzy sets: the ENNreg model." IEEE Transactions on Fuzzy Systems 31.10 (2023): 3690-3699.
>
> ---
> Table 9. Hyperparameter study on the effect of λ on the KIRC dataset. Metrics are reported as mean values.
> | λ   | C-index | IBS    | IBLL   |
> |-----|---------|--------|--------|
> | 0.1 | **0.7386** | 0.2574 | 0.7368 |
> | 0.3 | 0.7306  | 0.2078 | 0.5947 |
> | 0.5 | 0.7261  | 0.1788 | 0.5262 |
> | 0.7 | 0.7268  | 0.1615 | 0.4885 |
> | 0.9 | 0.7267  | **0.1556** | **0.4828** |
>
> ---
>
> Table 10. Hyperparameter study on the effect of λ on the UCEC dataset. Metrics are reported as mean values.
> | λ   | C-index | IBS    | IBLL   |
> |-----|---------|--------|--------|
> | 0.1 | **0.7190** | 0.2440 | 0.6764 |
> | 0.3 | 0.7097  | 0.1778 | 0.5185 |
> | 0.5 | 0.7099  | 0.1440 | 0.4432 |
> | 0.7 | 0.7104  | 0.1203 | 0.3916 |
> | 0.9 | 0.7108  | **0.1113** | **0.3718** |

---

> > ### Comment · Reviewer_MZaC · 2025-11-27
> >
> > I appreciate the response. The added experiments, especially the hyperparameter studies and the component alignment clarification, do indeed alleviate many of my initial concerns.
> >
> > However, my biggest concern that remains is related to the selection and justification of the hyperparameter λ. Your new sensitivity analysis, e.g., Table 9 and 10, demonstrates that both IBS and IBLL significantly improve, as one moves λ away from the selected value of 0.1. This suggests that model calibration is very sensitive to λ and that the chosen value yields suboptimal performances w.r.t. this key metric. The "conservative" justification feels post-hoc and not formally incorporated into the model's optimization.
> >
> > While I appreciate the additional clarification provided on component alignment and computational cost, the sensitivity of the model's calibration to this key hyperparameter remains a significant weakness that tempers this paper's claims of providing well-calibrated uncertainty estimates. Hence, I will keep my score.

---

> > > ### Author Response · Authors · 2025-11-27
> > > **Official Response to Reviewer MZaC**
> > >
> > > Dear Reviewer MZaC,
> > >
> > > Thanks for detailing your concerns. We believe there may be a misunderstanding regarding the role of the hyperparameter $\lambda$ and its relationship to performance calibration. We clarify these points below in four parts. **We trust that these explanations directly address your comments and hope they will prompt you to reconsider your evaluation.**
> > >
> > > **(1)  The primary function of $\lambda$ is to balance predictive accuracy and uncertainty.**
> > > A larger $\lambda$ reduces the model’s tolerance to uncertainty and places more emphasis on calibration; a smaller $\lambda$ does the opposite. However, it is important to clarify that the source of well-calibrated uncertainty in DPsurv does not come from $\lambda$ itself. Rather, calibration is fundamentally enabled by the proposed Gaussian random fuzzy numbers (GRFNs) and the dual-prototype evidential framework, which jointly model both aleatory and epistemic uncertainty.
> > >
> > > In evidential theory, the quantity $Bel_{\tilde Y}([x,y])$ can be interpreted as the degree of belief that **supports** the hypothesis $[x, y]$, while $Pl_{\tilde Y}([x,y])$ measures the degree of belief that **does not contradict** the hypothesis $[x, y]$. These correspond respectively to the expected necessity and expected possibility of the event taking a value within $[x, y]$.
> > >
> > > Therefore, in  Equation 12, $\lambda$ provides an explicit mechanism to adjust the model’s tolerance to uncertainty. This explains why increasing $\lambda$ in Tables 9 and 10 improves calibration at the cost of a marginal drop in C-index. Importantly, $\lambda$ offers a user-defined (or potentially data-driven) means of controlling the trade-off between accuracy and uncertainty, **while conventional probabilistic models do not inherently provide.**
> > >
> > >
> > > **(2) $\lambda$ is not formally incorporated into the model's optimization.**
> > > To simplify the model, we fixed $\lambda$ at a user-defined value of 0.1. While $\lambda$ can indeed be incorporated into the optimization to provide more flexible control between predictive accuracy and uncertainty, this requires larger-scale datasets for stable tuning. We appreciate this suggestion and will consider it in future work as more extensive data become available.
> > >
> > > **(3) The “conservative’’ behavior is not post hoc.**
> > > By “conservative survival estimate,’’ we refer to the model’s ability to produce more cautious predictions than probabilistic approaches because it explicitly models both aleatory and epistemic uncertainty, as demonstrated in paper [1]. The calibration results in Figure 3 of our paper further support this claim: the proportions of $\alpha$-level ($\alpha \in \{0.1,\ldots,0.9\}$) BPIs (Belief Prediction Intervals) consistently lie above the diagonal and exceed those of PPIs (Probability Prediction Intervals), indicating that BPIs are indeed more conservative than PPIs.
> > >
> > > **(4) “well-calibrated’’ uncertainty is a relative, not absolute, description.**
> > > We emphasize that “well-calibrated’’ is used in a relative sense. Across all experiments, **DPsurv consistently achieves substantially better calibration (IBS, IBLL) than strong baselines**. Although increasing $\lambda$ can further enhance calibration, our chosen setting provides a principled balance: it delivers well-calibrated uncertainty estimates while maintaining strong discriminative performance. This balance is essential for survival prediction, where both calibration and discrimination are required for clinically reliable models.
> > >
> > > **Furthermore, we would like to highlight that the sensitivity study of $\lambda$ is not the main focus relative to the core contributions of our work**, as its behavior has already been extensively investigated in recent studies such as [1–5]. The main contributions of our paper are:
> > > (1) an end-to-end interpretable dual-prototype architecture, and
> > > (2) explicit modeling of aleatory and epistemic uncertainty via belief–plausibility intervals.
> > > **Both contributions remain fully validated and unaffected by this hyperparameter.**
> > >
> > > [1] Denoeux, Thierry. *Quantifying prediction uncertainty in regression using random fuzzy sets: the ENNreg model.* IEEE Transactions on Fuzzy Systems 31.10 (2023): 3690–3699.
> > > [2] Denoeux, Thierry. *Reasoning with fuzzy and uncertain evidence using epistemic random fuzzy sets: General framework and practical models.* Fuzzy Sets and Systems 453 (2023): 1–36.
> > > [3] Denoeux, Thierry. *Parametric families of continuous belief functions based on generalized Gaussian random fuzzy numbers.* Fuzzy Sets and Systems 471 (2023): 108679.
> > > [4] Huang, Ling, et al. *EsurvFusion: An evidential multimodal survival fusion model based on epistemic random fuzzy sets.* IEEE Transactions on Fuzzy Systems (2025).
> > > [5] Huang, Ling, et al. *Evidential time-to-event prediction with calibrated uncertainty quantification.* International Journal of Approximate Reasoning 181 (2025): 109403.
> > >
> > > Best Regards,
> > >
> > > The Authors

---

> ### Author Response · Authors · 2025-11-21
> **Official Response to Reviewer MZaC(2/n)**
>
> **W3:** Computational complexity and scalability.
>
> **AW3:** Thank you for the comment on computational complexity and scalability. Although GMM is performed on each slide, its cost is significantly reduced by operating in the GMM embeddings rather than directly on millions of raw patches.
>
> To provide an empirical comparison, we report computation time on BLCA (437 WSIs), the smallest cohort, and BRCA (1111 WSIs), the largest and highest-resolution cohort. This allows us to assess scalability under both low- and high-N settings. Importantly, the reported DPsurv runtime includes all stages of the pipeline, including GMM-based WSI embedding extraction, component-prototype clustering, and evidential fusion training.
>
> As shown in Table 11, DPsurv exhibits a relatively small computational overhead despite including additional steps. The efficiency primarily comes from GMM aggregation, which reduces each high-resolution WSI to a compact mixture representation, keeping downstream computations inexpensive. Furthermore, the similar runtime scaling between BLCA and BRCA demonstrates that the framework can be applied to larger cohorts and higher-resolution WSIs without substantial increases in cost.
>
> We have included the results in Appendix H of the revised paper.
>
> ---
>
> Table 11. Training time comparison (in seconds) across methods on BLCA and BRCA datasets
> | **Method** | **BLCA** | **BRCA** |
> | ---------- | -------- | -------- |
> | ABMIL      | 956.91   | 1747.28  |
> | TransMIL   | 1449.01  | 3109.04  |
> | BayesMIL   | 959.23   | 2224.95  |
> | Panther    | 291.78   | 381.22   |
> | DPsurv     | 550.86   | 972.25   |

---

> ### Author Response · Authors · 2025-11-21
> **Official Response to Reviewer MZaC(3/n)**
>
> **W4:** Lack of experiments on hyperparameter studies.
>
> **AW4:** Thanks for your suggestion on hyperparameter studies. We have conducted a comprehensive sensitivity analysis covering all major hyperparameters in DPsurv, including the number of prototypes ($C, K$), trade-off terms ($\alpha$, $\lambda$), and regularization strengths ($R_1, R_2$). The results are summarized in Tables 12 and Table 13. Across a broad range of values and two independent datasets, the performance fluctuations remain small, indicating that DPsurv is robust and does not require extensive hyperparameter tuning.
>
> We have included the results in Appendix G of the revised paper.
>
> ---
>
> Table 12. Hyperparameter study on the KIRC dataset. Metrics are reported as mean values.
>
> | Parameter | Setting | C-index | IBS | IBLL |
> |----------|---------|---------|---------|---------|
> | **K** | 2 | 0.7271 | **0.2068** | **0.6049** |
> |        | 3 | 0.7289 | 0.2219 | 0.6388 |
> |        | 4 | **0.7386** | 0.2574 | 0.7368 |
> |        | 5 | 0.7166 | 0.2800 | 0.8013 |
> |        | 6 | 0.7217 | 0.2925 | 0.8436 |
> | **C** | 8  | 0.7287 | 0.2759 | 0.8136 |
> |        | 16 | **0.7386** | 0.2574 | 0.7368 |
> |        | 32 | 0.7176 | **0.2375** | **0.6838** |
> | **λ** | 0.1 | **0.7386** | 0.2574 | 0.7368 |
> |        | 0.3 | 0.7306 | 0.2078 | 0.5947 |
> |        | 0.5 | 0.7261 | 0.1788 | 0.5262 |
> |        | 0.7 | 0.7268 | 0.1615 | 0.4885 |
> |        | 0.9 | 0.7267 | **0.1556** | **0.4828** |
> | **α** | 0.1 | 0.7300 | **0.2202** | **0.6337** |
> |        | 0.3 | 0.7295 | 0.2316 | 0.6621 |
> |        | 0.5 | **0.7386** | 0.2574 | 0.7368 |
> |        | 0.7 | 0.7267 | 0.2918 | 0.8381 |
> |        | 0.9 | 0.7120 | 0.4079 | 1.2030 |
> | **R₁** | 0     | 0.7310 | **0.2479** | **0.7013** |
> |         | 0.005 | 0.7273 | 0.2520 | 0.7131 |
> |         | 0.01  | **0.7386** | 0.2574 | 0.7368 |
> |         | 0.1   | 0.7309 | 0.2502 | 0.7096 |
> | **R₂** | 0     | 0.7291 | **0.2450** | **0.6973** |
> |         | 0.005 | 0.7287 | 0.2513 | 0.7125 |
> |         | 0.01  | **0.7386** | 0.2574 | 0.7368 |
> |         | 0.1   | 0.7333 | 0.2473 | 0.7015 |
>
> ---
>
> Table 13. Hyperparameter study on the UCEC dataset. Metrics are reported as mean values.
>
> | Parameter | Setting | C-index | IBS | IBLL |
> |----------|---------|---------|---------|---------|
> | **K** | 2 | **0.7491** | **0.1366** | **0.4257** |
> |        | 3 | 0.7190 | 0.2440 | 0.6764 |
> |        | 4 | 0.7106 | 0.2158 | 0.6098 |
> |        | 5 | 0.7063 | 0.2271 | 0.6354 |
> |        | 6 | 0.7309 | 0.2398 | 0.6727 |
> | **C** | 8  | **0.7293** | 0.2698 | 0.7545 |
> |        | 16 | 0.7190 | 0.2440 | 0.6764 |
> |        | 32 | 0.7185 | **0.2375** | **0.6838** |
> | **λ** | 0.1 | **0.7190** | 0.2440 | 0.6764 |
> |        | 0.3 | 0.7097 | 0.1778 | 0.5185 |
> |        | 0.5 | 0.7099 | 0.1440 | 0.4432 |
> |        | 0.7 | 0.7104 | 0.1203 | 0.3916 |
> |        | 0.9 | 0.7108 | **0.1113** | **0.3718** |
> | **α** | 0.1 | 0.6927 | **0.1839** | **0.5348** |
> |        | 0.3 | 0.7022 | 0.2077 | 0.5882 |
> |        | 0.5 | **0.7190** | 0.2440 | 0.6764 |
> |        | 0.7 | 0.7093 | 0.2886 | 0.8263 |
> |        | 0.9 | 0.7140 | 0.3840 | 1.1160 |
> | **R₁** | 0     | 0.7076 | **0.2287** | **0.6362** |
> |         | 0.005 | 0.7072 | 0.2284 | 0.6351 |
> |         | 0.01  | **0.7190** | 0.2440 | 0.6764 |
> |         | 0.1   | 0.7049 | 0.2251 | 0.6293 |
> | **R₂** | 0     | 0.7062 | **0.2261** | **0.6317** |
> |         | 0.005 | 0.7066 | 0.2270 | 0.6346 |
> |         | 0.01  | **0.7190** | 0.2440 | 0.6764 |
> |         | 0.1   | 0.7072 | 0.2276 | 0.6353 |

---

> ### Author Response · Authors · 2025-11-21
> **Official Response to Reviewer MZaC(4/n)**
>
> **W5:** Minor issues.
>
> **AW5:** Thank you for pointing out the unclear explanantion of Eq.~(3)--(4). We have revised the manuscript and provide the full closed-form expression now. We clarify that $\varphi_i$ takes as input the $d$-dimensional patch embeddings and the corresponding GMM responsibilities, and outputs the component-wise aggregated statistics.
>
> Each of the $C$ components produces $(2d+1)$ values, consisting of the component mean, component variance, and the mixture weight. Therefore, the final per-slide representation has dimension $\mathbb{R}^{C \times (2d+1)}$.
>
> This will be explicitly stated in the revised version to avoid confusion.
>
> _Thank you once again for your detailed and constructive review. We hope that our clarifications and additional analyses have addressed your concerns. We look forward to any further feedback and are happy to provide additional information if needed._
>
> Best Regards,
>
> The Authors

---

### Official Review · Reviewer_AfEP · 2025-10-25

**Soundness:** 2
**Presentation:** 3
**Contribution:** 3
**Rating:** 4
**Confidence:** 3

**Summary:**

This paper focuses on the problems of interpretability, uncertainty, and tissue heterogeneity in survival prediction with whole slide images. The authors proposed DPsurv, a dual-prototype whole-slide image evidential fusion network that outputs uncertainty-aware survival intervals, while enabling interpretation of predictions through patch prototype assignment maps, component prototypes, and component-wise relative risk aggregation. The comparison of other baseline methods shows the good performance of the proposed method.

**Strengths:**

1. The paper is well-written and stated the problems logically, well motivated, and the figures are clear and easy to understand.
2. The comparison experiment is extensive.

**Weaknesses:**

1. While the paper presents an interesting framework, a major shortcoming is the absence of a systematic ablation study. DPsurv introduces several novel components, including the dual-prototype embedding, component-wise evidential fusion, and mixture evidential loss. However, the contribution of each module is not quantitatively isolated. The current comparisons against baseline models only provide indirect evidence of effectiveness.
2. Sensitivity analyses on the number of prototypes (C, K) and the trade-off parameters (α, λ...) are missing.
3. Limited novelty. The paper's novelty only lies in combining prototype-based representation learning with evidential reasoning.

**Questions:**

1. Could the authors provide a clear ablation study isolating the effects of (a) the dual-prototype embedding, (b) evidential fusion, and (c) uncertainty modeling? Currently, their individual contributions remain unclear.
2. Have the authors evaluated how sensitive the results are to the trade-off parameters (α, λ...) and the number of prototypes (C, K)?
3. What is the computational overhead of the dual-prototype evidential fusion compared to simpler MIL models? Can the framework scale to larger cohorts or higher-resolution WSIs?
4. Can you explain more about your main contribution compared to CVPR 2024 paper [1].

[1] Song, Andrew H., et al. "Morphological prototyping for unsupervised slide representation learning in computational pathology." Proceedings of the IEEE/CVF Conference on Computer Vision and Pattern Recognition. 2024.

If you can address my concern, I would like to raise my rating.

---

> ### Author Response · Authors · 2025-11-21
> **Official Response to Reviewer AfEP(1/n)**
>
> **W1 & Q1:** The absence of a systematic ablation study.
>
> **AW1 & AQ1:**  Thank you for the helpful suggestion. To directly address points (a)–(c), we performed a step-wise ablation where each module is toggled, as summarized in Table 5. Specifically, (a) the dual-prototype embedding corresponds to the **CP** column, (b) evidential fusion corresponds to the **EM** column, and (c) uncertainty modeling is implemented via the **GRFN** evidential architecture.
>
> From the table we can see, enabling GRFN primarily improves calibration, adding component prototypes and the evidence-mixture module brings further gains by modeling structured histological patterns, and the full model yields the best discriminative performance without degrading calibration.
>
> We have included ablation study in Appendix F of the revised paper.
>
> ---
>
> Table 5. Ablation study on the DPsurv architecture across five TCGA datasets. Metrics are reported as mean values.  **GRFN** indicates whether the evidential GRFN architecture is enabled.  **CP** denotes whether component-level prototype aggregation is used.  **EM** refers to whether evidence mixture fusion is applied.
>
> | **GMM** | **GRFN** | **CP** | **EM** | **C-index** | **IBS** | **IBLL** |
> |--------|----------|-----------------------------|---------------------------|-------------|---------|----------|
> | ✓ | ✗ | ✗ | ✗ | 0.6584 | 0.5851 | 1.5743 |
> | ✓ | ✓ | ✗ | ✗ | 0.6634 | 0.3305 | 0.8937 |
> | ✓ | ✓ | ✓ | ✗ | 0.5380 | 0.3003 | 0.9529 |
> | ✓ | ✓ | ✗ | ✓ | 0.6445 | **0.2197** | **0.7203** |
> | ✓ | ✓ | ✓ | ✓ | **0.6850** | 0.2880 | 0.8620 |

---

> ### Author Response · Authors · 2025-11-21
> **Official Response to Reviewer AfEP(2/n)**
>
> **W2 & Q2:** Sensitivity analyses are missing.
>
> **AW2 & AQ2:** Thanks for your comments on the sensitivity of the trade-off parameters ($\alpha$,$\lambda$) and the number of prototypes (C, K). In our submitted manuscript, we follow paper [1] that sets $\lambda$ as 0.1 and paper [2,3] that sets parameters $\alpha$ as 0.5, and the number of prototypes C was set as 16 following the paper [3], the number of component prototypes K was set based on empirical data. We thank you for the suggestion and have now added a sensitivity analysis to study the robustness of our proposal with those parameters on two cancer datasets, KIRC and UCEC. The results shown in Tables 6 and Table 7 indicate that performance varies slightly across a broad range of these settings in 5-fold cross-site validation, suggesting that DPsurv is robust and does not require heavy hyperparameter tuning. For example, the C-index for KIRC remains within a relatively tight band of approximately 0.72–0.74 across many choices of $K$, $C$, and $\lambda$, and UCEC similarly stays around 0.70–0.72.
>
> We have included the results in Appendix G of the revised paper.
>
> [1] Denoeux, Thierry. "Quantifying prediction uncertainty in regression using random fuzzy sets: the ENNreg model." IEEE Transactions on Fuzzy Systems 31.10 (2023): 3690-3699.
>
> [2] Song, Andrew H., et al. "Multimodal prototyping for cancer survival prediction." Proceedings of the 41st International Conference on Machine Learning. 2024.
>
> [3] Song, Andrew H., et al. "Morphological prototyping for unsupervised slide representation learning in computational pathology." Proceedings of the IEEE/CVF Conference on Computer Vision and Pattern Recognition. 2024.
> }
>
> ---
>
> Table 6. Hyperparameter study on the KIRC dataset. Metrics are reported as mean values.
>
> | Parameter | Setting | C-index | IBS | IBLL |
> |----------|---------|---------|---------|---------|
> | **K** | 2 | 0.7271 | **0.2068** | **0.6049** |
> |        | 3 | 0.7289 | 0.2219 | 0.6388 |
> |        | 4 | **0.7386** | 0.2574 | 0.7368 |
> |        | 5 | 0.7166 | 0.2800 | 0.8013 |
> |        | 6 | 0.7217 | 0.2925 | 0.8436 |
> | **C** | 8  | 0.7287 | 0.2759 | 0.8136 |
> |        | 16 | **0.7386** | 0.2574 | 0.7368 |
> |        | 32 | 0.7176 | **0.2375** | **0.6838** |
> | **λ** | 0.1 | **0.7386** | 0.2574 | 0.7368 |
> |        | 0.3 | 0.7306 | 0.2078 | 0.5947 |
> |        | 0.5 | 0.7261 | 0.1788 | 0.5262 |
> |        | 0.7 | 0.7268 | 0.1615 | 0.4885 |
> |        | 0.9 | 0.7267 | **0.1556** | **0.4828** |
> | **α** | 0.1 | 0.7300 | **0.2202** | **0.6337** |
> |        | 0.3 | 0.7295 | 0.2316 | 0.6621 |
> |        | 0.5 | **0.7386** | 0.2574 | 0.7368 |
> |        | 0.7 | 0.7267 | 0.2918 | 0.8381 |
> |        | 0.9 | 0.7120 | 0.4079 | 1.2030 |
> | **R₁** | 0     | 0.7310 | **0.2479** | **0.7013** |
> |         | 0.005 | 0.7273 | 0.2520 | 0.7131 |
> |         | 0.01  | **0.7386** | 0.2574 | 0.7368 |
> |         | 0.1   | 0.7309 | 0.2502 | 0.7096 |
> | **R₂** | 0     | 0.7291 | **0.2450** | **0.6973** |
> |         | 0.005 | 0.7287 | 0.2513 | 0.7125 |
> |         | 0.01  | **0.7386** | 0.2574 | 0.7368 |
> |         | 0.1   | 0.7333 | 0.2473 | 0.7015 |
>
> ---
>
> Table 7. Hyperparameter study on the UCEC dataset. Metrics are reported as mean values.
>
> | Parameter | Setting | C-index | IBS | IBLL |
> |----------|---------|---------|---------|---------|
> | **K** | 2 | **0.7491** | **0.1366** | **0.4257** |
> |        | 3 | 0.7190 | 0.2440 | 0.6764 |
> |        | 4 | 0.7106 | 0.2158 | 0.6098 |
> |        | 5 | 0.7063 | 0.2271 | 0.6354 |
> |        | 6 | 0.7309 | 0.2398 | 0.6727 |
> | **C** | 8  | **0.7293** | 0.2698 | 0.7545 |
> |        | 16 | 0.7190 | 0.2440 | 0.6764 |
> |        | 32 | 0.7185 | **0.2375** | **0.6838** |
> | **λ** | 0.1 | **0.7190** | 0.2440 | 0.6764 |
> |        | 0.3 | 0.7097 | 0.1778 | 0.5185 |
> |        | 0.5 | 0.7099 | 0.1440 | 0.4432 |
> |        | 0.7 | 0.7104 | 0.1203 | 0.3916 |
> |        | 0.9 | 0.7108 | **0.1113** | **0.3718** |
> | **α** | 0.1 | 0.6927 | **0.1839** | **0.5348** |
> |        | 0.3 | 0.7022 | 0.2077 | 0.5882 |
> |        | 0.5 | **0.7190** | 0.2440 | 0.6764 |
> |        | 0.7 | 0.7093 | 0.2886 | 0.8263 |
> |        | 0.9 | 0.7140 | 0.3840 | 1.1160 |
> | **R₁** | 0     | 0.7076 | **0.2287** | **0.6362** |
> |         | 0.005 | 0.7072 | 0.2284 | 0.6351 |
> |         | 0.01  | **0.7190** | 0.2440 | 0.6764 |
> |         | 0.1   | 0.7049 | 0.2251 | 0.6293 |
> | **R₂** | 0     | 0.7062 | **0.2261** | **0.6317** |
> |         | 0.005 | 0.7066 | 0.2270 | 0.6346 |
> |         | 0.01  | **0.7190** | 0.2440 | 0.6764 |
> |         | 0.1   | 0.7072 | 0.2276 | 0.6353 |

---

> ### Author Response · Authors · 2025-11-21
> **Official Response to Reviewer AfEP(3/n)**
>
> **Q3:** What is the computational overhead of the dual-prototype evidential fusion compared to simpler MIL models? Can the framework scale to larger cohorts or higher-resolution WSIs?
>
> **AQ3:** Thanks for the questions. We have evaluated the computation cost of our model compared with simpler MIL models (The results are shown in Table 8). To provide a representative comparison, we report results on BLCA (437 WSIs), the smallest cohort, and BRCA (1111 WSIs), the largest cohort in our study. Both datasets contain very high–resolution WSIs, enabling us to assess computational behavior under both low- and high-volume settings.
>
> **Scalability.**
> As shown in Table 8, the overall computational overhead of DPsurv remains relatively small. This efficiency is largely attributed to the GMM aggregation, which compresses each WSI into a compact set of mixture features and keeps the downstream evidential model lightweight. Moreover, the similar runtime pattern observed for BLCA and BRCA suggests that DPsurv scales well to larger cohorts and can accommodate higher-resolution WSIs.
>
> We have included the results in Appendix H of the revised paper.
>
> ---
>
> Table 8. Training time comparison (in seconds) across methods on BLCA and BRCA datasets
> | **Method** | **BLCA** | **BRCA** |
> | ---------- | -------- | -------- |
> | ABMIL      | 956.91   | 1747.28  |
> | TransMIL   | 1449.01  | 3109.04  |
> | BayesMIL   | 959.23   | 2224.95  |
> | Panther    | 291.78   | 381.22   |
> | DPsurv     | 550.86   | 972.25   |

---

> ### Author Response · Authors · 2025-11-21
> **Official Response to Reviewer AfEP(4/n)**
>
> **W3 & Q4:** Can you explain more about your main contribution？
>
> **AW3 & AQ4:** Thank you for the comment. Structurally, the CVPR 2024 paper (Panther) is used in our work only as an encoder to extract slide embeddings. Task-wise, Panther focuses on representation learning, whereas our paper focuses on survival prediction, which requires modeling survival functions with uncertainty quantification and interpretability. We detail the distinctions below.
>
> **1. Interpretability for survival prediction.**
> Panther provides an interpretable morphological composition of a WSI, for example, the distribution of tumor, stroma and necrosis, but it does not explain how these different phenotypes affect survival risk. DPsurv uses a dual-prototype design that enables end-to-end interpretability in the following three parts:
>
> (1) GMM provides the morphological distribution of a WSI;
>
> (2) Component prototypes model the risk characteristics of different phenotypes;
>
> (3) Evidence mixture aggregation links region-specific risk evidence to the final survival prediction.
>
> To the best of our knowledge, this is the first work in WSI survival analysis to explicitly model and aggregate the survival evidence of different histological phenotypes within a slide.
>
> **2. Uncertainty quantification.**
> Survival outcomes are often censored, making uncertainty quantification essential. Panther does not study uncertainty.
> DPsurv incorporates the GRFN evidential framework, which estimates both aleatoric and epistemic uncertainty and yields calibrated risk intervals suitable for censored data.
>
> **3. Novel architectural contribution recognized by reviewers.**
> Several reviewers highlighted the novelty and relevance of our approach. Reviewer **McuC** noted a novel hierarchical architecture for interpretability and principled integration of uncertainty quantification. Reviewer **MZaC** stated that uncertainty quantification and interpretable reasoning for WSI survival are both relevant and challenging. Reviewer **Uv5K** emphasized that modeling feature phenotypes and survival evidence separately with a dual-prototype architecture is a novel concept.
>
> _Thank you once again for your detailed and constructive review. We hope that our clarifications and additional analyses have addressed your concerns. We look forward to any further feedback and are happy to provide additional information if needed._
>
> Best Regards,
>
> The Authors

---

> > ### Comment · Reviewer_AfEP · 2025-11-24
> > **Concerns Addressed**
> >
> > Thank you for your detailed explanation. Now I would like to raise my rating to 6.

---

> > > ### Author Response · Authors · 2025-11-24
> > > **Appreciation for Reviewer AfEP’s Follow-up**
> > >
> > > Dear Reviewer AfEP,
> > >
> > > **We are sincerely grateful for your follow-up and for raising your rating.**
> > >
> > > Thank you for your constructive suggestions on the ablation study, hyperparameter analysis, and computational cost. Your feedback has been invaluable in helping us further refine and strengthen our manuscript. We are pleased that our rebuttal was able to address your concerns.
> > >
> > > _We truly appreciate your time and thoughtful evaluation throughout the review process._
> > >
> > > Best regards,
> > >
> > > The Authors

---

### Official Review · Reviewer_McuC · 2025-11-01

**Soundness:** 3
**Presentation:** 2
**Contribution:** 3
**Rating:** 6
**Confidence:** 5

**Summary:**

The authors present DPsurv, a novel framework for survival prediction from whole-slide images that uniquely combines interpretability with uncertainty quantification. The core contribution is a dual-prototype architecture: it first employs unsupervised Gaussian Mixture Models (GMM) to encode morphological heterogeneity into **patch prototypes**, then maps these representations to a survival prediction using a second layer of **component prototypes** grounded in evidential theory. This approach explicitly models both aleatoric and epistemic uncertainty, enabling the generation of well-calibrated Belief Prediction Intervals (BPIs). Experiments on five TCGA cohorts demonstrate state-of-the-art performance in both discrimination (C-index) and calibration (IBS), while offering multi-level interpretability from patch features to prognostic reasoning.

**Strengths:**

The paper makes several notable contributions to the field of computational pathology, particularly in its approach to building more transparent and reliable survival prediction models.

**1. A Novel Hierarchical Architecture for Interpretability:**
The primary strength of this work is its introduction of a novel dual-prototype architecture that mirrors a pathologist's diagnostic reasoning. By first using unsupervised patch prototypes to categorize morphological features and then employing a second layer of component prototypes to link these features to prognostic outcomes, the model creates a structured and semantically meaningful reasoning path. This hierarchical design is a creative combination of existing ideas (GMMs, prototype learning) and represents a significant conceptual advance over flatter, end-to-end deep learning models, offering a clear pathway toward feature-level and case-level interpretability.

**2. Principled Integration of Uncertainty Quantification:**
A key merit of the paper is its rigorous and principled approach to uncertainty modeling. Instead of relying on ad-hoc or purely probabilistic methods, the authors ground their framework in evidential theory, using Gaussian Random Fuzzy Numbers to explicitly disentangle aleatoric (data-inherent) and epistemic (model-based) uncertainty. This is a technically sophisticated choice that is well-suited to the problem, as it allows the model to report not just a prediction, but also its own confidence in that prediction. The experimental results, particularly the superior calibration performance measured by IBS, suggest that this approach is highly effective.

**3. Strong Empirical Performance on Diverse Datasets:**
The quality and significance of the proposed method are substantiated by strong empirical results. The authors conduct a thorough evaluation across five different cancer types from the TCGA database, which represent a diverse set of challenges in terms of tissue morphology and data characteristics. The fact that DPsurv consistently achieves state-of-the-art or competitive performance in both discrimination (C-index) and, crucially, calibration (IBS) against a range of relevant baselines validates the effectiveness of the overall framework. This robust performance across multiple datasets underscores the potential of this approach to be a generalizable tool for survival analysis.

**Weaknesses:**

Despite its promising contributions, the paper exhibits several significant weaknesses that temper its conclusions and limit the reproducibility and reliability of the proposed framework. The following points outline specific areas that require substantial improvement.

**1. Lack of Ablation Studies to Justify Architectural Complexity:**
The central weakness of this paper is the absence of a thorough ablation study to validate its complex, multi-stage architecture. The DPsurv model combines several sophisticated components (GMMs, a dual-prototype structure, and an evidential fusion mechanism), but their individual contributions to the final performance are not disentangled. To justify this complexity, the authors should provide experiments that answer the following critical questions:
*   **Is the dual-prototype structure essential?** What is the performance drop if the second layer of component prototypes is removed, and the GMM-derived features are directly mapped to the evidential output layer? This would clarify the true value of the evidence-fusion stage.
*   **How does the evidential framework compare to alternatives?** The claim of superior uncertainty quantification would be much stronger if DPsurv's evidential components were replaced with more standard uncertainty techniques like Monte Carlo Dropout or Deep Ensembles, followed by a direct comparison of their IBS scores and calibration plots. This is crucial for demonstrating that the benefits stem from the evidential framework itself, not just from having an uncertainty-aware component in general.
*   **What is the impact of the regularization terms?** The contribution of the `R1` and `R2` regularization terms in the mixture evidential loss (Eq. 15) is unclear. An analysis showing performance with and without these terms would clarify their role in stabilizing the training and improving calibration.
Without these ablations, it is difficult to ascertain whether the model's success is truly due to its intricate design or if a simpler model could achieve comparable results.

**2. Insufficient Analysis of Hyperparameter Sensitivity:**
The model's reliability is questionable due to the large number of crucial hyperparameters and the lack of a sensitivity analysis. Key parameters include the number of patch prototypes (`C`), the number of component prototypes (`K`), and the loss trade-off parameter (`alpha`).
*   **The choice of `K` lacks empirical support.** The authors provide a qualitative, knowledge-driven rationale for selecting `K` for each dataset (Appendix B.4). While insightful, this is not a substitute for empirical validation. The paper would be significantly strengthened by including a table or plot showing how performance (e.g., C-index and IBS) varies with different choices of `K` for at least one or two datasets. This would demonstrate the robustness of the model to this critical parameter.

**3. Subjective and Limited Evaluation of Interpretability:**
The paper claims interpretability as a core contribution, yet its evaluation is purely qualitative and based on a single case study (Figure 2). This is insufficient to support a general claim of interpretability.
*   **Lack of objective metrics.** The evaluation could be strengthened by introducing more objective, even if semi-quantitative, assessments. For instance, the authors could conduct a user study where multiple pathologists are asked to rate the clinical relevance and coherence of the prototypes and assignment maps across a larger set of randomly selected cases. Their inter-rater agreement could then be reported.
*   **Risk of cherry-picking.** The presented case in Figure 2 may be a "best-case" example. To mitigate concerns of selection bias, the authors should present additional, randomly sampled case studies (perhaps in the appendix) to demonstrate the consistency of the model's interpretable outputs. A model's interpretability is only valuable if it is reliable and consistent, not just illustrative in one favorable instance.

**4. Clarity of Exposition Could Be Improved for Broader Accessibility:**
While technically detailed, the paper's exposition could be made more accessible to a broader audience. The reliance on dense terminology from evidential theory (e.g., Gaussian Random Fuzzy Numbers, belief/plausibility functions) without sufficient high-level intuition can make the core mechanism difficult to grasp for readers not already expert in this specific niche. Incorporating more analogies or simplified step-by-step explanations of how evidence is mathematically fused would significantly improve the paper's clarity and impact, helping to bridge the gap between the complex theory and its practical application.

**Questions:**

Thank you for this interesting and thought-provoking work. Your approach of integrating a dual-prototype structure with evidential theory is highly novel. To better understand the contributions and limitations of your framework, I have a few questions and suggestions that I hope will be helpful for the discussion phase.

1.  **On the Necessity of the Dual-Prototype Architecture:** The multi-stage design is intricate. To better understand the contribution of each stage, could you please provide an ablation study? Specifically, what would the performance be if the second layer of component prototypes was ablated, and the GMM-derived component summary (the output of the first stage) was fed directly into a simpler evidential regression head? This would help clarify the exact performance gain attributable to the similarity-based evidence fusion mechanism.

2.  **On the Choice of the Evidential Framework:** The use of GRFNs is central to your claim of superior uncertainty quantification. To place this contribution in the context of more standard methods, could you provide a direct comparison against a strong non-evidential uncertainty baseline, such as Deep Ensembles, trained on the same architecture? A comparison of the IBS scores and calibration plots (like in Figure 3) would be particularly insightful to empirically demonstrate the advantages of the evidential approach over these well-established alternatives.

3.  **On Hyperparameter Sensitivity, Particularly for `K`:** The number of component prototypes, `K`, seems to be a critical and manually tuned parameter. While the clinical rationale provided in the appendix is appreciated, the paper would be much more convincing if this choice were supported by empirical data. Could you please provide a sensitivity analysis for `K` on at least one or two of the datasets? For example, a plot showing how the C-index and IBS vary as `K` is changed (e.g., from 2 to 6) would provide valuable insight into the model's robustness and help justify the values you selected.

4.  **On the Evaluation of Interpretability:** The interpretability claims are primarily supported by a single, albeit detailed, case study in Figure 2. To strengthen this claim and address potential concerns of cherry-picking:
    *   Could you provide additional examples of the model's interpretable outputs on other, randomly selected cases, perhaps in the appendix? Seeing the model's reasoning on more average or ambiguous cases would be very helpful.
    *   Have you considered any form of more objective evaluation? For instance, even a small-scale study where pathologists assess the clinical coherence of the discovered prototypes across a larger set of cases could provide more robust evidence than a single illustrative example.

5.  **On the Intuition behind Evidence Fusion:** For readers who are not experts in evidential theory, the mathematical fusion of GRFNs in Equation 8 can be difficult to grasp intuitively. Would it be possible to provide a simplified, high-level walkthrough of this step? For example, a toy example with two component prototypes and a target component, showing how their respective `μ`, `σ²`, and `h` values are combined, would greatly improve the accessibility of your core mechanism.

**Details Of Ethics Concerns:**

N/A.

---

> ### Author Response · Authors · 2025-11-21
> **Official Response to Reviewer McuC(1/n)**
>
> **W1.1 & Q1:** The multi-stage design is intricate. To better understand the contribution of each stage, could you please provide an ablation study?
>
> **AW1 & AQ1:** Thank you for your comments. To better understand the contribution and effectiveness of the Dual-Prototype Architecture, we present (i) an ablation study that evaluates its impact on predictive performance, and (ii) an interpretability analysis showing how the second layer of component prototypes enhance model interpretability.
>
> **(i) Ablation Study.**
> We decomposed the Dual-Prototype Architecture into four ablation parts: GMM, GRFN, Component Prototype, and Evidence Mixture.  The results in Table 1 show a clear gain of each module: GRFN substantially improves calibration, while component prototypes and evidence mixture provide additional gains. The full model achieves the highest discriminative performance without compromising calibration quality.
>
> **(ii) Interpretability Analysis.**
> The design of the second layer of component prototypes also enhances decision interpretability by modeling component-level risk evidence. For example, the second layer of component prototypes may learn several tumor subtypes, each associated with a distinct level of risk evidence. For a new whole-slide image, the model infers the risk of its tumor regions by measuring their similarity to these learned component prototypes. The closer a region is to a high-risk prototype, the higher the resulting evidential risk, enabling transparent, component-level interpretation.
>
> We have included ablation study in Appendix F of the revised paper.
>
> ---
>
> Table 1. Ablation study on the DPsurv architecture across five TCGA datasets. Metrics are reported as mean values.  **GRFN** indicates whether the evidential GRFN architecture is enabled.  **CP** denotes whether component-level prototype aggregation is used.  **EM** refers to whether evidence mixture fusion is applied.
>
> | **GMM** | **GRFN** | **CP** | **EM** | **C-index** | **IBS** | **IBLL** |
> |--------|----------|-----------------------------|---------------------------|-------------|---------|----------|
> | ✓ | ✗ | ✗ | ✗ | 0.6584 | 0.5851 | 1.5743 |
> | ✓ | ✓ | ✗ | ✗ | 0.6634 | 0.3305 | 0.8937 |
> | ✓ | ✓ | ✓ | ✗ | 0.5380 | 0.3003 | 0.9529 |
> | ✓ | ✓ | ✗ | ✓ | 0.6445 | **0.2197** | **0.7203** |
> | ✓ | ✓ | ✓ | ✓ | **0.6850** | 0.2880 | 0.8620 |

---

> ### Author Response · Authors · 2025-11-21
> **Official Response to Reviewer McuC(2/n)**
>
> **W1.2 & Q2:** How does the evidential framework compare to alternatives?
>
> **AW1.2 & AQ2:** Thank you for your insightful suggestion. We have conducted additional experimental comparisons between our method and two strong non-evidential uncertainty baselines, MC Dropout and Deep Ensemble.  The comparison results in Table 2 show that our method achieves superior predictive performance across all three evaluation metrics.  To make this clearer, we summarize the key differences between GRFN-based uncertainty modeling and traditional non-evidential baselines below.
>
> **(1) Disentangling epistemic and aleatoric uncertainty.**
> MC Dropout and Deep Ensemble estimate uncertainty via sampling or model ensembling, but they do not distinguish between epistemic and aleatoric uncertainty.  In contrast, GRFN explicitly models both components at the evidence level.
>
> **(2) Predictive calibration through uncertainty weighting.**
> While MC Dropout and Deep Ensemble only quantify uncertainty, GRFN not only estimates uncertainty but also leverages it to optimize model training.  In particular, evidences with higher uncertainty automatically receive lower weights during aggregation, enabling more robust and interpretable predictions.
>
> We have included the results and calibration plots in Appendix K of the revised paper.
>
> ---
>
> Table 2. Comparison of uncertainty modeling approaches on survival prediction across five TCGA datasets. Metrics are reported as mean values.
>
> | Method         | C-index | IBS     | IBLL    |
> |----------------|---------|---------|---------|
> | MC Dropout     | 0.6454  | 0.7743  | 2.8195  |
> | Deep Ensemble  | 0.6483  | 0.8033  | 3.5054  |
> | **Ours**       | **0.6850** | **0.2880** | **0.8620** |

---

> ### Author Response · Authors · 2025-11-21
> **Official Response to Reviewer McuC(3/n)**
>
> **W1.3:** What is the impact of the regularization terms?
>
> **AW1.3:** Thank you for your comments. We conducted additional experiments on both the KIRC and UCEC datasets to examine the impact of the regularization terms. The comparison results in Table 3 show that introducing proper regularization improves discriminative performance without substantially sacrificing calibration.
>
> We have included the results in Appendix G of the revised paper.
>
> ---
>
> Table 3. Impact of regularization terms R₁ and R₂ on survival prediction (KIRC & UCEC).
> Metrics are reported as mean values.
>
> | Dataset | Reg. | Value | C-index | IBS | IBLL |
> |--------|------|--------|---------|---------|---------|
> | **KIRC** | R₁ | 0     | 0.7310 | **0.2479** | **0.7013** |
> |         |     | 0.005 | 0.7273 | 0.2520 | 0.7131 |
> |         |     | 0.01  | **0.7386** | 0.2574 | 0.7368 |
> |         |     | 0.1   | 0.7309 | 0.2502 | 0.7096 |
> |         | R₂ | 0     | 0.7291 | **0.2450** | **0.6973** |
> |         |     | 0.005 | 0.7287 | 0.2513 | 0.7125 |
> |         |     | 0.01  | **0.7386** | 0.2574 | 0.7368 |
> |         |     | 0.1   | 0.7333 | 0.2473 | 0.7015 |
> | **UCEC** | R₁ | 0     | 0.7076 | **0.2287** | **0.6362** |
> |         |     | 0.005 | 0.7072 | 0.2284 | 0.6351 |
> |         |     | 0.01  | **0.7190** | 0.2440 | 0.6764 |
> |         |     | 0.1   | 0.7049 | 0.2251 | 0.6293 |
> |         | R₂ | 0     | 0.7062 | **0.2261** | **0.6317** |
> |         |     | 0.005 | 0.7066 | 0.2270 | 0.6346 |
> |         |     | 0.01  | **0.7190** | 0.2440 | 0.6764 |
> |         |     | 0.1   | 0.7072 | 0.2276 | 0.6353 |

---

> ### Author Response · Authors · 2025-11-21
> **Official Response to Reviewer McuC(4/n)**
>
> **W2 & Q3:** Insufficient analysis of hyperparameter sensitivity.
>
> **AW2 & AQ3:** Thank you for your comments and suggestions regarding the hyperparameter analysis.  The number of component prototypes K is indeed an important hyperparameter. While K can in principle be tuned using a hold-out validation set, this approach is often unreliable for cancer survival datasets because of their limited sample size and high censoring rate. Splitting data into additional validation folds may further exacerbate distributional imbalance and make the model highly sensitive to sampling noise. Therefore, in our submitted manuscript, we determined K based on empirical observations.
>
> To provide a clearer assessment of its impact, we conducted a sensitivity analysis of K ranging from 2 to 6 under a 5-fold cross-site validation protocol, ensuring that WSIs from the same acquisition center never appear in both training and testing folds. Results on the KIRC and UCEC datasets (Tables 3 & Table 4) show that the performance of DPsurv remains within a reasonable range across different values of K. This indicates that the method is stable and robust, without requiring finely tuned prototype counts.
>
> Moreover, we carried out a full sensitivity analysis over all hyperparameters, and the results consistently show that DPsurv remains stable and robust across a wide range of settings.
>
> We have included the results in Appendix G of the revised paper.
>
> ---
>
> Table 3. Hyperparameter study on the KIRC dataset. Metrics are reported as mean values.
>
> | Parameter | Setting | C-index | IBS | IBLL |
> |----------|---------|---------|---------|---------|
> | **K** | 2 | 0.7271 | **0.2068** | **0.6049** |
> |        | 3 | 0.7289 | 0.2219 | 0.6388 |
> |        | 4 | **0.7386** | 0.2574 | 0.7368 |
> |        | 5 | 0.7166 | 0.2800 | 0.8013 |
> |        | 6 | 0.7217 | 0.2925 | 0.8436 |
> | **C** | 8  | 0.7287 | 0.2759 | 0.8136 |
> |        | 16 | **0.7386** | 0.2574 | 0.7368 |
> |        | 32 | 0.7176 | **0.2375** | **0.6838** |
> | **λ** | 0.1 | **0.7386** | 0.2574 | 0.7368 |
> |        | 0.3 | 0.7306 | 0.2078 | 0.5947 |
> |        | 0.5 | 0.7261 | 0.1788 | 0.5262 |
> |        | 0.7 | 0.7268 | 0.1615 | 0.4885 |
> |        | 0.9 | 0.7267 | **0.1556** | **0.4828** |
> | **α** | 0.1 | 0.7300 | **0.2202** | **0.6337** |
> |        | 0.3 | 0.7295 | 0.2316 | 0.6621 |
> |        | 0.5 | **0.7386** | 0.2574 | 0.7368 |
> |        | 0.7 | 0.7267 | 0.2918 | 0.8381 |
> |        | 0.9 | 0.7120 | 0.4079 | 1.2030 |
> | **R₁** | 0     | 0.7310 | **0.2479** | **0.7013** |
> |         | 0.005 | 0.7273 | 0.2520 | 0.7131 |
> |         | 0.01  | **0.7386** | 0.2574 | 0.7368 |
> |         | 0.1   | 0.7309 | 0.2502 | 0.7096 |
> | **R₂** | 0     | 0.7291 | **0.2450** | **0.6973** |
> |         | 0.005 | 0.7287 | 0.2513 | 0.7125 |
> |         | 0.01  | **0.7386** | 0.2574 | 0.7368 |
> |         | 0.1   | 0.7333 | 0.2473 | 0.7015 |
>
> ---
>
> Table 4. Hyperparameter study on the UCEC dataset. Metrics are reported as mean values.
>
> | Parameter | Setting | C-index | IBS | IBLL |
> |----------|---------|---------|---------|---------|
> | **K** | 2 | **0.7491** | **0.1366** | **0.4257** |
> |        | 3 | 0.7190 | 0.2440 | 0.6764 |
> |        | 4 | 0.7106 | 0.2158 | 0.6098 |
> |        | 5 | 0.7063 | 0.2271 | 0.6354 |
> |        | 6 | 0.7309 | 0.2398 | 0.6727 |
> | **C** | 8  | **0.7293** | 0.2698 | 0.7545 |
> |        | 16 | 0.7190 | 0.2440 | 0.6764 |
> |        | 32 | 0.7185 | **0.2375** | **0.6838** |
> | **λ** | 0.1 | **0.7190** | 0.2440 | 0.6764 |
> |        | 0.3 | 0.7097 | 0.1778 | 0.5185 |
> |        | 0.5 | 0.7099 | 0.1440 | 0.4432 |
> |        | 0.7 | 0.7104 | 0.1203 | 0.3916 |
> |        | 0.9 | 0.7108 | **0.1113** | **0.3718** |
> | **α** | 0.1 | 0.6927 | **0.1839** | **0.5348** |
> |        | 0.3 | 0.7022 | 0.2077 | 0.5882 |
> |        | 0.5 | **0.7190** | 0.2440 | 0.6764 |
> |        | 0.7 | 0.7093 | 0.2886 | 0.8263 |
> |        | 0.9 | 0.7140 | 0.3840 | 1.1160 |
> | **R₁** | 0     | 0.7076 | **0.2287** | **0.6362** |
> |         | 0.005 | 0.7072 | 0.2284 | 0.6351 |
> |         | 0.01  | **0.7190** | 0.2440 | 0.6764 |
> |         | 0.1   | 0.7049 | 0.2251 | 0.6293 |
> | **R₂** | 0     | 0.7062 | **0.2261** | **0.6317** |
> |         | 0.005 | 0.7066 | 0.2270 | 0.6346 |
> |         | 0.01  | **0.7190** | 0.2440 | 0.6764 |
> |         | 0.1   | 0.7072 | 0.2276 | 0.6353 |

---

> ### Author Response · Authors · 2025-11-21
> **Official Response to Reviewer McuC(5/n)**
>
> **W3 & Q4:** Subjective and limited evaluation of interpretability.
>
> **AW3 & AQ4:**  Thank you for the insightful comments. We address both points as follows.
>
> **1. Additional randomly selected interpretability examples.**
> To reduce concerns of cherry-picking, we include two additional randomly selected whole-slide examples in the appendix. Each example presents the assignment map and relative risk visualization, showing that the interpretability patterns observed in Figure 2 are consistent across typical and more ambiguous cases.
>
> **2. More objective evaluation of interpretability.**
> We further conduct a small-scale clinical coherence assessment. We randomly sample a subset of slides, generate their assignment map and relative risk visualization using DPsurv, and ask pathologists to rate the coherence between model-generated interpretations and their clinical insights on a scale from 1 to 10. The average score is reported as an objective interpretability metric, providing more robust evidence than a single illustrative example. Additional details are provided in the Appendix I of the revised paper.
>
> ---
>
> **W4 & Q5:** Clarity of exposition could be improved for broader accessibility.
>
> **AW4 & AQ5:**  Thank you for the helpful suggestion. We agree that a high-level and intuitive walkthrough can greatly improve accessibility for readers who are less familiar with evidential theory. To address this, we have added a simplified toy example in the appendix. This example considers two component prototypes and one target component, and illustrates step-by-step how their corresponding $(\mu, \sigma^2, h)$ values are fused following Equation 8. The toy example provides a concrete demonstration of how evidential risk contributions from different components are aggregated, making the core fusion mechanism easier to grasp. We have included the example in Appendix J of the revised paper.
>
> _Thank you once again for your detailed and constructive review. We hope that our clarifications and additional analyses have addressed your concerns. We look forward to any further feedback and are happy to provide additional information if needed._
>
> Best Regards,
>
> The Authors

---

### Author Response · Authors · 2025-11-24
**Revised Manuscript Uploaded**

Dear Reviewers,

We would like to inform you that the revised version of our manuscript has been successfully uploaded.

In this revision, we have addressed all reviewer comments, and the corresponding changes have been incorporated as follows:

- **Ablation study** has been added on page 22, lines 1150–1155.
- **Hyperparameter study** has been added on page 22, lines 1173–1177.
- **Computational cost** analysis has been added on page 22, lines 1181–1187, and page 25, lines 1296–1298.
- Two additional **interpretability examples** and our proposed **objective evaluation criteria** have been added on pages 25–26.
- A **toy example for clearer walkthrough** has been added on page 27, lines 1405–1426.
- IBS results and calibration plots comparing to strong **uncertainty quantification baselines** have been included on page 27, lines 1431–1457.
- A **clearer component definition** has been added on page 4, lines 202–204.
- The definition of the **function $\phi$** and **clarification of final dimensionality** have been added on page 17, lines 872–880.

All modifications have been highlighted in **blue** in the revised manuscript for easy reference.

We are grateful for your valuable feedback, which has significantly strengthened our manuscript. If there are any additional points or concerns, we would be glad to address them promptly.

Best Regards,

The Authors

---

### Author Response · Authors · 2025-11-27
**Request for Further Clarification (Final Week)**

Dear Reviewer,

We sincerely thank you again for your constructive feedback.

As the rebuttal period has now entered its **final week**, we would be grateful if you could let us know whether any further clarification or additional analysis from our side would be helpful for your assessment.
We remain very happy to provide any further details promptly.

Best regards,

The Authors

---

### Author Response · Authors · 2025-12-01
**Summary of the rebuttal discussion**

Dear Reviewers, AC, SAC, and PC,

Thank you very much for your time and insightful feedback.
For your convenience, we provide here a concise summary of the rebuttal discussion.

We are grateful for the positive comments we received, including **the significance of the target problem** (AfEP, MZaC, Uv5K), **the novelty of our hierarchical architecture** (McuC, MZaC, Uv5K), **the strength of our uncertainty quantification and calibration approach** (McuC, MZaC), **the solid performance across diverse datasets** (McuC, AfEP, MZaC, Uv5K), and **the interpretability that offers meaningful clinical insights** (MZaC, Uv5K). These comments are truly encouraging to us.

**We would also like to express our appreciation to Reviewer AfEP for the follow-up during the discussion period and for reconsidering the evaluation by raising the score from 4 to 6.** The reviewer’s reassessment, along with the thoughtful questions raised during the discussion period, helped us further improve the manuscript. Below we summarize the key concerns and our corresponding responses, based on both the initial reviews and the reviewers’ follow-up.

**Summary of Reviewers’ Questions**
 - Aablation study (McuC, AfEP, Uv5K)
 - Comprehensive hyperparameter studies (McuC, AfEP, MZaC, Uv5K)
 - Computational cost and scalability (MZaC, Uv5K)
 - More interpretability examples (McuC)
 - Toy example to clarify the workflow (McuC)
 - Comparison with uncertainty quantification baselines (McuC, Uv5K)
 - Clarification on the novelty relative to existing work (AfEP)
 - Clarification of certain definitions (MZaC)
 - Minor issues (MZaC)

**Our Responses to the Reviewers’ Questions**

 - Ablation study

    Following the suggestions from McuC, AfEP, and Uv5K, we conducted a comprehensive ablation study demonstrating the effectiveness of our dual-prototype architecture. The ablation results have been added on page 22, lines 1150–1155.**This concern is addressed based on reviewer AfEP’s response.**

 - Hyperparameter study

    Following the suggestions from McuC, AfEP, MZaC, and Uv5K, we conducted a comprehensive hyperparameter study on the KIRC and UCEC datasets. The results show that DPsurv remains robust across a wide range of settings. The study has been added on page 22, lines 1173–1177. **This concern is addressed based on reviewer AfEP’s and MZaC’s responses.**

 - Computational cost analysis

    Following the suggestions from MZaC and Uv5K, we performed a detailed computational cost analysis covering all major components of DPsurv, including prototype initialization, GMM fitting, and model training. The results show that our method is computationally efficient compared to baseline models (ABMIL, TransMIL, and BayesMIL). The analysis has been added on page 22, lines 1181–1187, and page 25, lines 1296–1298. **This concern is addressed based on reviewer MZaC’s response.**

 - Additional interpretability examples

    Following the suggestions from McuC, we added two additional interpretability examples together with our proposed objective evaluation criteria on pages 25–26. We further strengthened this section by incorporating evaluations from a pathologist, confirming that the component-wise risk and assignment maps provide clinically meaningful insights.


 - Toy example for clearer walkthrough

    Following the suggestions from McuC, we added a toy example to provide a clearer step-by-step walkthrough of our method. This addition can be found on page 27, lines 1405–1426.

 - Comparison with uncertainty quantification baselines

    Following the suggestions from McuC and Uv5K, we added IBS results and calibration plots comparing our method with strong uncertainty-quantification baselines (MC-Dropout and Deep Ensemble). These results have been included on page 27, lines 1431–1457.

 - Novelty clarification

    Following the suggestions from AfEP, we provided additional clarification in the rebuttal regarding the novelty of our method relative to existing work. **This concern is addressed based on reviewer AfEP’s response.**

 - Clearer definition of components

    Following the suggestions from MZaC, we added a clearer definition of the components on page 4, lines 202–204. **This concern is addressed based on reviewer MZaC’s response.**

 - Minor issues

    Following the suggestions from MZaC, we addressed all minor issues and wording clarifications on page 17, lines 872–880. **This concern is addressed based on reviewer MZaC’s response.**

We sincerely thank the Reviewers, AC, SAC, and PC for their effort and thoughtful feedback throughout the ICLR 2026 review process.

Best Regards,

The Authors

---

### Meta-Review · Area_Chair_w3ZD · 2025-12-29

**Summary:**

This work presents DPsurv, a framework for survival prediction from WSIs. Four reviewers have carefully reviewed this paper and provided valuable comments. One reviewer gave a rating of 6 (marginally above the acceptance threshold), and three reviewers gave a rating of 4 (marginally below the acceptance threshold). The overall rating is below the acceptance bar.

The reviewers are all concerned about the complexity of the framework, pointing out the lack of ablation studies, computation costs evaluation, and hyperparameter analysis. The authors respond by adding experiments in the rebuttal. However, the inherent complexity problem still exists. For example, this framework combines multiple modules, including a foundation model, GMMs, two levels of prototypes, and the evidential framework of GRFNs, which is a typical “a+b+c” work. In the limitation part (Appendix 1121-1130), the authors said this framework is limited by the selection of hyperparameters. While in the rebuttal, the authors overturn their own conclusion, saying that the DPsurv does not rely on hyperparameter selection (response to Reviewer Uv5K). This contradiction may make it difficult to trust the presented results. Considering there are multiple modules that rely on multiple manual hyperparameters, I think this is not a trivial problem

While the paper has some strengths, they are not strong enough to merit presentation at a top-tier conference like ICLR. Given the overall negative rating and unresolved concerns, I am sorry to recommend rejection.

**Reviewer Concerns:**

The complexity of the framework is not solved.

**Reviewer Scores:**

Reviewer AfEP may improve the rating. Reviewers MZaC and Uv5K may keep the negative ratings.

---

### Decision · Program_Chairs · 2026-01-26

Reject